# Double Machine Learning Density Estimation for Local Treatment Effects with Instruments

**Yonghan Jung**
Purdue University
jung222@purdue.edu

**Jin Tian**
Iowa State University
jtian@iastate.edu

**Elias Bareinboim**
Columbia University
eb@cs.columbia.edu

## Abstract

Local treatment effects are a common quantity found throughout the empirical sciences that measure the treatment effect among those who comply with what they are assigned. Most of the literature is focused on estimating the average of such quantity, which is called the "*local average treatment effect (LATE)*" [31]). In this work, we study how to estimate the density of the *local treatment effect*, which is naturally more informative than its average. Specifically, we develop two families of methods for this task, namely, kernel-smoothing and model-based approaches. The kernel-smoothing-based approach estimates the density through some smooth kernel functions. The model-based approach estimates the density by projecting it onto a finite-dimensional density class. For both approaches, we derive the corresponding double/debiased machine learning-based estimators [13]. We further study the asymptotic convergence rates of the estimators and show that they are robust to the biases in nuisance function estimation. The use of the proposed methods is illustrated through both synthetic and a real dataset called 401(k).

## 1 Introduction

Controlled experimentation is one powerful tool used throughout the empirical sciences to infer the effect of a certain treatment on a given outcome. The idea is to randomize the treatment assignment so as to neutralize the effect of unobserved confounders. However, in some practical settings, it may be challenging to ascertain that individuals who are selected for treatment will follow their recommendations. Issues of non-compliance and unmeasured confounding are quite common and lead to the non-identification of treatment effects in many real-world cases [29, 50, 32, 56].

An approach known as instrumental variables (IVs) has been proposed to try to circumvent this issue [68]. The idea is to find a set of variables (possibly singleton) that are not the target of the analysis by itself but that will help to control for the unobserved confounding between the treatment and the outcome. In particular, IVs are special variables that (i) are correlated with the treatment, (ii) do not directly influence the outcome, and (iii) are not affected by certain unmeasured confounders. For concreteness, consider a study of the effect of 401(k) participation ($X$) on the distribution of net financial assets ($Y$) [2]. This setting is represented in the causal graph in Fig. 1. Note that a dashed-bidirected arrow exists between $X$ and $Y$, which in graphical language represents unobserved confounding affecting both $X$ and $Y$. The variable $Z$ in this model represents the eligibility of 401(k). We note that $Z$ qualifies as an instrument in this case – (i) it does affect the participation of 401(k) ($X$) and (ii) has no direct influence on the net financial asset ($Y$), (iii) is not affected by unmeasured confounders between $X$ and $Y$. The variable $W$ represents observed covariates (e.g., gender, age, ethnicity, income, family size).

We are interested in the particular setting where only individuals who were offered the treatment may have access to it [31]. For instance, in the case of 401(k) participation ($X = 1$), only eligible

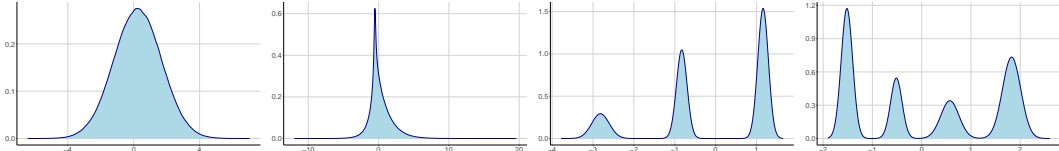

Figure 2: Densities of outcome $Y$ among compliers under the treatment $X = 1$. All densities have a mean 0 and a variance 2.

individuals ($Z = 1$) would be allowed to join the program. This assumption is known in the literature as *monotonicity*, which rules out the possibility that any units would respond contrary to the instrument. Under monotonicity, the causal effect in the subpopulation whose actual treatment $X$ coincides with the assigned treatment $Z$ (called *compliers*) is identifiable [31, 2]. The average treatment effect (ATE) for the compliers is called 'Local ATE' (LATE) (or Complier average causal effects, CACE) [31].

The most common quantification of these effects in IV settings found in practice is the average (e.g., LATE). The average is certainly an informative summary; however, it may fail to capture significant differences in the causal distributions of the outcome. For instance, consider Fig. 2 that shows the densities of outcomes $Y$ under treatments $X = 1$ among compliers which are generated from samples drawn from four synthetic data generating processes represented by the IV graph in Fig. 1 (further discussed in Sec. 5). All of the four distributions have the same mean 0 and variance 2. However, the difference in the LTE distributions is self-evident.

Most of the prior work on quantifying distributions of treatment effects focuses on estimating cumulative distribution functions (CDFs) or quantiles, and little attention has been given to estimating densities (refer to Sec. 1.1 for further comparison). As a complement to CDFs, densities have various advantages, including a more interpretable visualization of the distribution and generative capability of producing samples. One challenge with estimating densities is that while CDFs are pathwise-differentiable and enjoy $\sqrt{n}$-rate estimators ($n$ is the size of data), densities are not (i.e., they are *non-regular*), and therefore possess no influence functions nor $\sqrt{n}$-rate estimators without approximations [7, Ch. 3].

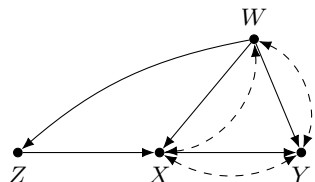

Figure 1: A causal graph for the IV setting. Bidirected arrows encode unmeasured confounders.

In this paper, our goal is to provide methods to estimate densities of local treatment effects in IV settings under the monotonicity assumption. We develop two families of methods for this task based on *kernel-smoothing* and *model-based* approximations. The former smooths the density by convolution with a kernel function; the latter projects the density onto a finite-dimensional density class based on a distributional distance measure. For both approaches, we construct double/debiased machine learning (DML) style density estimators [43, 54, 52, 70, 13]. We analyze the asymptotic convergence properties of the estimators, showing that they can converge fast (i.e., $\sqrt{n}$-rate) even when nuisance estimates converge slowly (e.g., $n^{-1/4}$ rate) (a property called '*debiasedness*'[1]). We illustrate the proposed methods on synthetic and real data.

## 1.1 Related work

Our work touches different areas, which we will discuss next.

**Double/Debiased Machine Learning (DML) [13]-based causal effect estimators.** The DML framework has been adapted for estimating the average causal effect under the setting where the *back-door criterion* [50, Sec. 3.3.1] (also known as ignorability [57]) holds (e.g., [12, 19]). Recently, DML-based causal effect estimators have been developed for any identifiable causal functionals in a given causal graph and equivalence class thereof [33, 34].

---

[1]Also known as '*nonparametric doubly robust*' [37] or '*rate doubly robust*' [59].

**Local average & quantile treatment effect.** The formal identification results for LATE under the monotonicity assumption in IV settings were developed by [31, 3]. Building on these results, semiparametric estimation for LATE has received remarkable attention [2, 60, 23, 62, 48], including robust LATE estimators that achieve debiasedness [47, 40, 38, 64]. As shown in Fig. 2, however, the average is sometimes insufficient to capture the effects of the treatment on the distributions of outcomes. To address this issue, the problem of estimating quantiles or CDFs has taken attention. A common approach to estimate quantiles or CDFs is based on the LATE estimation. Since the expectation of $\mathbb{1}_{Y \leq y}(Y)$, an indicator that outcome $Y$ falls short of threshold $y$, reduces to the CDF (i.e., replacing $Y$ in LATE with $\mathbb{1}_{Y \leq y}(Y)$), estimators for the LATE can be used to estimate quantiles or CDFs [1, 2, 15, 24, 16, 30, 45, 18, 69].

**Non-regular target estimand.** An estimand that possesses no influence functions nor $\sqrt{n}$-rate estimators is called '*non-regular*'. Densities are an example of non-regular target estimands [7, Chap. 3]. One can approximate a non-regular target with a smooth one such that an influence function and $\sqrt{n}$-rate estimators can be derived. Two broadly used approaches are *kernel-smoothing-based* (e.g., [52, 6, 42, 19, 35]) and *model-based* (e.g., [46, 52, 21, 41, 40, 39]).

**Causal density estimation.** There is limited literature on estimating the density of treatment effects. Most of the results assume that the ignorability/backdoor admissibility holds [55, 49]. [22] used the kernel-smoothing technique to estimate the density of a treatment effect, and [42] provided a kernel-smoothing-based density estimator that achieves doubly robustness and debiasedness building on top of the work in [53]. Recently, [39] investigated a model-based approach and developed estimators that achieve debiasedness properties. Under the IV setting, [10] provided a local polynomial regression-based density estimator for local treatment effects; We are not aware of any work studying debiased density estimators. As mentioned, this paper investigates both kernel-smoothing and model-based approaches for estimating local treatment effects under IV settings and develops DML-style density estimators for both.

## 2 LTE Estimation – Problem setup

In our analysis, each variable is represented with a capital letter $(X)$ and its realized value with a small letter $(x)$. For a discrete (e.g., binary) random variable $X$, we use $\mathbb{1}_x(X)$ to represent the indicator function such that $\mathbb{1}_x(X) = 1$ if $X = x$; $\mathbb{1}_x(X) = 0$ otherwise. For a continuous variable $X$ with a probability density $p(x)$ of a distribution $P$ and a function $f(x)$, $\mathbb{E}_P[f(X)] \equiv \int_{\mathcal{X}} f(x)p(x) \, d[x]$ where $\mathcal{X}$ is the domain for $X$, and $\|f(X)\| \equiv \sqrt{\mathbb{E}_P[(f(X))^2]}$. $\widehat{f}$ is said to converge to $f$ at rate $r_n$ if $\|\widehat{f}(x) - f(x)\| = O_P(1/r_n)$. For a dataset $\mathcal{D} = \{V_i\}_{i=1}^n$, we use $\mathbb{E}_{\mathcal{D}}[f(V)] \equiv (1/n) \sum_{i=1}^n f(V_i)$ to denote the empirical mean of $f(V)$ with $\mathcal{D}$.

**Structural Causal Models (SCMs).** We use the language of SCMs as our basic semantic and inferential framework [50, 4]. An SCM $\mathcal{M}$ is a quadruple $\mathcal{M} = \langle U, V, P(U), F \rangle$ where $U$ is a set of exogenous (latent) variables following a joint distribution $P(u)$, and $V$ is a set of endogenous (observable) variables whose values are determined by functions $F = \{f_{V_i}\}_{V_i \in V}$ such that $V_i \leftarrow f_{V_i}(pa_i, u_i)$ where $PA_i \subseteq V$ and $U_i \subseteq U$. Each SCM $\mathcal{M}$ induces a distribution $P(v)$ and a causal graph $G = G(\mathcal{M})$ over $V$ in which there exists a directed edge from every variable in $PA_i$ to $V_i$ and dashed-bidirected arrows encode common latent variables (e.g., see Fig. 1). Within the structural semantics, performing an intervention and setting $X = x$ is represented through the do-operator, $do(X = x)$, which encodes the operation of replacing the original equations of $X$ (i.e., $f_X(pa_x, u_x)$) by the constant $x$ and induces a *submodel* $\mathcal{M}_x$ and an interventional distribution $P(v|do(x))$. For any variable $Y \in V$, the *potential response* $Y_x(u)$ is defined as the solution of $Y$ in the submodel $M_x$ given $U = u$, which induces a *counterfactual variable* $Y_x$.

**Local Treatment Effect (LTE) with IV.** We consider the IV setting represented by the causal graph $G$ in Fig. 1[2], where $Z$ is a binary instrument with domain $\{0, 1\}$, $X$ is a binary treatment with domain $\{0, 1\}$, and $Y$ is a (set of) continuous outcomes with bounded domain $\mathcal{Y} \subset \mathbb{R}^d$, and $W$ is a set of covariates (continuous, discrete, or mixed). $G$ satisfies the IV assumption that $Z$ has no direct influence on outcome $Y$ and is not affected by unmeasured confounders between $X$ and $Y$.

---

[2]It is common in the literature to define IV assumptions in terms of conditional independences among counterfactual [51, 9, 8, 2, 60, 47, 64], whose connection with the causal graph in Fig. 1 is discussed in Assumption A.1

The causal density $p(y_x)$ is not identifiable from the observed density $p(x, y, z, w)$ due to the unobserved confounders between $X$ and $Y$. However, the effect is possibly recovered for certain subpopulation under additional assumptions. Formally, a unit in the population is an always-taker if $X_{Z=1} = X_{Z=0} = 1$, a never-taker if $X_{Z=1} = X_{Z=0} = 0$, a *complier* if $X_{Z=1} = 1, X_{Z=0} = 0$, and a defier if $X_{Z=1} = 0, X_{Z=0} = 1$ [3, 2]. We will make the following assumptions based on literature.

**Assumption 1** (**Monotonicity**). *There are no defiers:* $X_{Z=1} \geq X_{Z=0}$.

**Assumption 2** (**Positivity**). $P(x|z, w) > 0, P(z|w) > 0$ *for any* $x, z, w$.

Let $C$ denote the event that a unit is a complier (i.e., a unit such that $X_{Z=0} = 0$ and $X_{Z=1} = 1$). For a given constant $a$ and a variable $X$, let $x^a$ denote the event $X = a$. The LTE $p(y_x|C)$ is identifiable under monotonicity and is given by [31, 2]:

$$p(y_x|C) = \frac{\mathbb{E}_P\left[p(y|x, z^x, W)P(x|z^x, W) - p(y|x, z^{1-x}, W)P(x|z^{1-x}, W)\right]}{\mathbb{E}_P\left[P(x^1|z^1, W) - P(x^1|z^0, W)\right]}, \tag{1}$$

where the expectation is over $W$. In this paper, we aim to estimate the LTE density $p(y_x|C)$ in Eq. (1). We will make the following mild assumption on some densities, popularly employed in density estimation literature (e.g., [44, 25, 27, 61, 26, 42]).

**Assumption 3.** *For any* $x, z, w, y$, *densities* $p(y|w, z, x)$, $p(y|z, x)$ *and* $p(y_x|C)$ *are bounded, and* $p(y_x|C)$ *is twice differentiable.*

**DML method**. Let $\psi \equiv \psi_{P'}$ denote a functional of an arbitrary distribution $P'$. We use $P$ to denote the true distribution such that $\mathcal{D} \sim P$. Let $\psi_0 \equiv \psi_P$ denote the true parameter to be estimated. To estimate $\psi_0$, DML-based estimators use a *Neyman Orthogonal score* $\varphi(V; \psi_0, \eta)$ (where $\eta \equiv \eta_{P'}$ is a set of nuisance parameters and $\eta_0 \equiv \eta_P$ denotes the true nuisances), a function such that $\mathbb{E}_P\left[\varphi(V; \psi_0, \eta_0)\right] = 0$, $(\partial/\partial\eta)|_{\eta=\eta_0}\mathbb{E}_P\left[\varphi(V; \psi_0, \eta)\right] = 0$. Given $\varphi$, an DML estimator is constructed using the *cross-fitting technique* as follows: For randomly split halves of $\mathcal{D}$ denoted $\{\mathcal{D}_0, \mathcal{D}_1\}$, let $\widehat{\eta}_p$ for $p \in \{0, 1\}$ denote the estimates for $\eta$ from $\mathcal{D}_p$. Let $T_p$ denote a solution such that $\mathbb{E}_{\mathcal{D}_{1-p}}\left[\varphi(V; T_p, \widehat{\eta}_p)\right] = o_P(N^{-1/2})$. Then, $T \equiv (T_0 + T_1)/2$ is an DML estimator [13, Def. 3.1]. In addition to being consistent, the estimator $T$ exhibits a robustness property called *debiasedness*: $T$ converges to $\psi_0$ in the root-$N$ rate even when $\widehat{\eta}$ converges to $\eta_0$ in slower $N^{-1/4}$ rate [13, Thm. 3.1]. A Neyman Orthogonal Score can be derived by adding $\psi$ to its *influence function* $\phi$ [14, Thm. 1]. An influence function of the functional $\psi_P$ is defined as a solution satisfying $\mathbb{E}_P\left[\phi\right] = 0$, $\mathbb{E}_P\left[\phi^2\right] < \infty$, and $(\partial/\partial t)\psi_{P_t}|_{t=0} = \mathbb{E}_P\left[\phi(V; \psi, \eta)S_t(V; t = 0)\right]$ where $P_t \equiv P(v)(1 + tg(v))$ for $t \in \mathbb{R}$ and any bounded mean-zero functions $g(\cdot)$ over $V$, and $S_t(v; t = 0) \equiv (\partial/\partial t)\log P_t(v)|_{t=0}$ [63, Chap. 25].

Due to space constraints, all the proofs are provided in Appendix B in suppl. material.

## 3 Kernel-smoothing-based approach

In this section, we develop a kernel-smoothing-based approach for estimating the LTE density. The kernel-smoothing technique approximates a non-pathwise-differentiable target estimand with a differentiable estimand by convoluting the density with a kernel function $K(y)$. Properties of the kernel function includes symmetry about the origin (i.e., $\int_{\mathcal{Y}} yK(y)\, d[y] = 0$), non-negativity $(0 < K(y) < \infty, \forall y \in \mathcal{Y})$, and integrates to 1 (i.e., $\int_{\mathcal{Y}} K(y)\, d[y] = 1$) [66, Chap. 4.2].

We consider a *product kernel* $K_{h,y}(y') \equiv h^{-d} \prod_{j=1}^{d} K((y_j - y'_j)/h)$ with given bandwidth $h \in \mathbb{R}$ and a fixed point $y = \{y_j\}_{j=1}^{d} \in \mathbb{R}^d$. We assume that the kernel of interest has a bounded second moment and norm: i.e., $\kappa_2(K) \equiv \int_{\mathcal{Y}} y^2 K(y)\, d[y] < \infty$ and $\|K(y)\| < \infty$ following [27, 61]. Example of kernels include Gaussian kernel: $K(u) = (1/\sqrt{2\pi})\exp\left(-u^2/2\right)$ for $u \in \mathbb{R}$, Epanechnikov kernel: $K(u) = (3/4)(1 - u^2)\mathbb{1}_{|u|\leq 1}(u)$, Quadratic kernel: $K(u) = (15/16)(1 - u^2)^2\mathbb{1}_{|u|\leq 1}(u)$, Cosine kernel: $k(u) = (\pi/4)\cos(\pi u/2)\mathbb{1}_{|u|\leq 1}(u)$, etc.

For convenience, we denote the target estimand by $\psi(y) \equiv p(y_x|C)$. In the kernel-smoothing-based approach, we will aim to estimate a kernel-smoothed approximation for $\psi(y)$ defined as follows:

$$\psi_h(y) \equiv \int_{\mathcal{Y}} \psi(y')K_{h,y}(y')\, d[y'] = \psi[K_{h,y}(Y)], \tag{2}$$

where $\psi[f(Y)]$ is an expectation of a function $f(Y)$ w.r.t. $\psi(y)$, which is specified as

$$\psi[f(Y)] \equiv \frac{\mathbb{E}_P\left[\mathbb{E}_P\left[f(Y)\mathbb{1}_x(X)|z^x, W\right] - \mathbb{E}_P\left[f(Y)\mathbb{1}_x(X)|z^{1-x}, W\right]\right]}{\mathbb{E}_P\left[P(x^1|z^1, W) - P(x^1|z^0, W)\right]}. \tag{3}$$

The second equality in Eq. (2) is by Eq. (1). For a target estimand $\psi[f(Y)]$, we will denote nuisances by $\pi_z(w) \equiv P(z|w)$, $\xi_x(z, w) \equiv P(x|z, w)$, and $\theta(x, z, w)[f(Y)] \equiv \mathbb{E}_P\left[f(Y)\mathbb{1}_x(X)|z, w\right]$, shortly $(\pi, \xi, \theta)$.

We aim to construct a DML estimator for the estimand $\psi_h$. Toward this goal, we will first derive a Neyman orthogonal score for $\psi_h$. Since a Neyman orthogonal score can be constructed based on *moment score functions* (a function of parameters such that its expectation is 0 at the true parameters) [14, Thm. 1], we start by defining the moment score function. Let

$$\psi^X \equiv \mathbb{E}_P\left[\xi_{x^1}(z^1, W) - \xi_{x^1}(z^0, W)\right], \tag{4}$$

$$\mathcal{V}_X(\{\pi, \xi\}) \equiv \frac{\mathbb{1}_{z^1}(Z) - \mathbb{1}_{z^0}(Z)}{\pi_Z(W)}\left\{\mathbb{1}_{x^1}(X) - \xi_{x^1}(Z, W)\right\} + \left\{\xi_{x^1}(z^1, W) - \xi_{x^1}(z^0, W)\right\}. \tag{5}$$

Then, the following is a moment score function for $\psi_h$:

$$m(\psi'; \psi_h) \equiv \frac{1}{\psi^X}\left(\psi_h - \psi'\right)\mathcal{V}_X, \tag{6}$$

where $\psi_h$ is given in Eq. (2) and $\psi'$ is an estimate of $\psi_h$.

Next, we derive an influence function for the moment score function $m(\psi'; \psi_h)$. We first define the following function: for a bounded function $f(Y) < \infty$, let

$$\psi^{YX}[f(Y)] \equiv \mathbb{E}_P\left[\theta(x, z^x, W)[f(Y)] - \theta(x, z^{1-x}, W)[f(Y)]\right], \tag{7}$$

$$\mathcal{V}_{YX}(\{\pi, \theta\})[f(Y)] \equiv \frac{\mathbb{1}_{z^x}(Z) - \mathbb{1}_{z^{1-x}}(Z)}{\pi_Z(W)}\left\{f(Y)\mathbb{1}_x(X) - \theta(x, Z, W)[f(Y)]\right\}$$
$$+ \left\{\theta(x, z^x, W)[f(Y)] - \theta(x, z^{1-x}, W)[f(Y)]\right\}, \tag{8}$$

and

$$\phi(\eta = \{\pi, \xi, \theta\}, \psi)[f(Y)] \equiv \frac{1}{\psi^X}\left(\mathcal{V}_{YX}(\{\pi, \theta\})[f(Y)] - \psi[f(Y)]\mathcal{V}_X(\{\pi, \xi\})\right), \tag{9}$$

where $\mathcal{V}_X$ is defined in Eq. (5). Then, the influence function for the expectation of the moment score function $m(\psi'; \psi_h)$ in Eq. (6) is given as follows:

**Lemma 1 (Influence function for $m(\psi'; \psi_h)$).** *Let $m(\psi'; \psi_h)$ be the score defined in Eq. (6). Then, the influence function for $\mathbb{E}_P\left[m(\psi'; \psi_h)\right]$, denoted $\phi_m$, is given by*

$$\phi_m(\eta = \{\pi, \xi, \theta\}, \psi) \equiv \phi(\eta, \psi)[K_{h,y}(Y)] \tag{10}$$

*where $\phi$ is in Eq. (9).*

For any score function (e.g., $m$ in Eq. (6)), its addition to the influence function of the expected score (e.g., $\phi_m$) is a Neyman orthogonal score[3] ([14, Thm.1], [13, Sec. 2.2.5]). Specifically,

**Lemma 2 (Neyman orthogonal score for $\psi_h$).** *Let $m(\psi'; \psi_h)$ be the score function in Eq. (6), and $\phi_m(\eta = \{\pi, \xi, \theta\}, \psi_h)$ be the influence function for $\mathbb{E}_P\left[m(\psi'; \psi_h)\right]$ given in Eq. (10). Then, a Neyman orthogonal score for $\psi_h$ is given as $\varphi(\psi'; \eta = \{\pi, \xi, \theta\}) \equiv m(\psi'; \psi_h) + \phi_m(\eta, \psi)$; Specifically,*

$$\varphi(\psi'; \eta = \{\pi, \xi, \theta\}) = \frac{1}{\psi^X}\left(\mathcal{V}_{YX}(\{\pi, \theta\})[K_{h,y}(Y)] - \psi'\mathcal{V}_X(\{\pi, \xi\})\right). \tag{11}$$

Given the Neyman orthogonal score $\varphi(\psi'; \eta)$, an estimate $\hat{\psi}_h$ satisfying $\mathbb{E}_{\mathcal{D}}\left[\varphi(\hat{\psi}_h; \hat{\eta} = \{\hat{\pi}, \hat{\xi}, \hat{\theta}\})\right] = o_P(n^{-1/2})$ gives a DML estimator. Specifically, we propose the following kernel-smoothing based estimator for the LTE density, named 'KLTE' (kernel-based estimator for LTE):

---

[3] A Neyman orthogonal score is a function $\phi$ satisfying $\mathbb{E}_P[\phi(\psi, \eta_0)] = 0$ and $\frac{\partial}{\partial\eta}\mathbb{E}_P[\phi(V; \psi, \eta)]|_{\eta=\eta_0} = 0$, where $\eta_0$ denotes the true nuisance [13, Def.2.2]. In words, a score function that is not sensitive to local errors in nuisance models.

**Definition 1** (**KLTE estimator for** $\psi_h$). *Let* $\varphi(\psi'; \eta = \{\pi, \xi, \theta\})$ *be the Neyman orthogonal score for* $\psi_h$ *given in Eq. (11). Let* $\{\mathcal{D}, \mathcal{D}'\}$ *denote the randomly split halves of the samples, where* $|\mathcal{D}| = |\mathcal{D}'| = n$. *Let* $\hat{\eta} = \{\hat{\pi}, \hat{\xi}, \hat{\theta}\}$ *denote the estimates for the nuisance* $\eta$ *using* $\mathcal{D}'$. *Then, the KLTE estimator for* $\psi_h(y)$ *for all* $y \in \mathcal{Y}$, *denoted* $\hat{\psi}_h(y)$, *is given by*

$$\hat{\psi}_h(y) \equiv \mathbb{E}_{\mathcal{D}}\left[\mathcal{V}_{YX}(\{\hat{\pi}, \hat{\theta}\})[K_{h,y}(Y)]\right] / \mathbb{E}_{\mathcal{D}}\left[\mathcal{V}_X(\{\hat{\pi}, \hat{\xi}\})\right], \tag{12}$$

*where* $\mathcal{V}_X$ *and* $\mathcal{V}_{YX}$ *are given in Eqs. (5,8), respectively.*

We will show that the KLTE is a DML estimator exhibiting debiasedness property. Detailed asymptotic properties are discussed next.

## 3.1 Asymptotic convergence

Now, we study the convergence rate of the estimator $\hat{\psi}_h(y)$. For any fixed $y \in \mathcal{Y}$, the error $\hat{\psi}_h(y) - \psi(y)$ will be analyzed in two folds: we will first analyze the error between the estimator in Eq. (12) and the smoothed estimand in Eq. (2) (i.e., $\hat{\psi}_h(y) - \psi_h(y)$), and then analyze the error between the smoothed estimand and the true estimand (i.e., $\psi_h(y) - \psi(y)$).

The following result gives the error analysis for $\hat{\psi}_h(y) - \psi_h(y)$:

**Lemma 3** (**Convergence rate of** $\hat{\psi}_h$ **to** $\psi_h$). *For any fixed* $y \in \mathcal{Y}$, *suppose the estimators for nuisances are consistent; i.e.,* $\|\nu - \hat{\nu}\| = o_P(1)$ *for* $\nu \in \eta = \{\pi, \xi, \theta\}$ *for all* $(w, z, x)$. *Suppose* $h < \infty$, *and* $nh^d \to \infty$ *as* $n \to \infty$. *Then,*

$$\hat{\psi}_h(y) - \psi_h(y) = O_P\left(1/\sqrt{nh^d} + R_2^k + 1/\sqrt{n}\right),$$

*where*

$$R_2^k \equiv \sum_z \|\hat{\pi}_z - \pi_z\| \left\{\left\|\hat{\theta}_z - \theta_z\right\| + \left\|\hat{\xi}_z - \xi_z\right\|\right\}, \tag{13}$$

*where* $\pi_z \equiv \pi_z(W)$, $\xi_z \equiv \xi_x(z, W)$ *and* $\theta_z \equiv \theta(x, z, W)[K_{h,y}(Y)]$.

The error analysis in Lemma. 3 implies the following:

**Corollary 1** (**Debiasedness property of** $\hat{\psi}_h$ **to** $\psi_h$). *If all nuisances* $\{\hat{\pi}, \hat{\xi}, \hat{\theta}\}$ *for any given* $(w, z, x, y)$ *converge at rate* $\{nh^d\}^{-1/4}$, *then the target estimator* $\hat{\psi}_h(y)$ *achieves* $\sqrt{nh^d}$*-rate convergence to* $\psi_h$.

We now analyze the gap between the smoothed estimand $\psi_h$ and the true estimand $\psi$; i.e., $\psi_h - \psi$:

**Lemma 4** ([66, Thm. 6.28]). *The following holds:*

$$\psi_h(y) - \psi(y) = B_y \equiv 0.5h^2\kappa_2(K)(\partial^2/\partial^2 y')|_{y'=y}\psi(y') + O(h^2). \tag{14}$$

Combining the results of Lemma. (3,4), we have the following result:

**Theorem 1** (**Convergence rate of** $\hat{\psi}_h$ **to** $\psi$). *For any fixed* $y \in \mathcal{Y}$, *suppose the estimators for nuisances are consistent; i.e.,* $\|\nu - \hat{\nu}\| = o_P(1)$ *for* $\nu \in \eta = \{\pi, \xi, \theta\}$ *for all* $(w, z, x)$. *Suppose* $h < \infty$, *and* $nh^d \to \infty$ *as* $n \to \infty$. *Then*

$$\hat{\psi}_h(y) - \psi(y) = O_P\left(1/\sqrt{nh^d} + R_2^k + 1/\sqrt{n}\right) + B_y, \tag{15}$$

*where* $B_y$ *is defined in Eq. (14), and* $R_2^k$ *is defined in Eq. (13).*

Thm. 1 implies that $\hat{\psi}_h(y)$ converges fast (see Corol. 1) to $\psi(y) + B_y$. A natural question is then how to choose the bandwidth $h$ that minimizes the gap in Eq. (15). The following provides a guideline in choosing the bandwidth $h$:

**Lemma 5** (**Data-adaptive bandwidth selection**). *The bandwidth* $h$ *that minimizes the error in Eq. (15) is* $h = O(n^{-1/(d+4)})$. *This choice of* $h$ *satisfies the assumption in Lemma 3 (i.e.,* $nh^d \to \infty$).

Recall that Corol. 1 states the debiasedness property of $\hat{\psi}_h$ to $\psi_h$ for any bandwidth $h$ satisfying $nh^d \to \infty$. With the choice of $h$ as in Lemma 5, $\hat{\psi}_h$ converges to $\psi$ with the debiasedness property preserved.

**Corollary 2** (**Debiasedness property of $\hat{\psi}_h$ to $\psi$**). *Let $h = O(n^{-1/(d+4)})$. If nuisances $\{\hat{\pi}, \hat{\xi}, \hat{\theta}\}$ converge at $\{nh^d\}^{-1/4}$ rate for any $(w, z, x, y)$, then the target estimator $\hat{\psi}_h(y)$ achieves $\sqrt{nh^d}$-rate convergence to $\psi$.*

So far, we have analyzed the error $\hat{\psi}_h(y) - \psi(y)$ pointwise for the fixed $y \in \mathcal{Y}$. To analyze the difference between the two densities $\hat{\psi}_h(y)$ and $\psi(y)$ for all $y \in \mathcal{Y}$, we consider the following divergence function of two densities:

**Definition 2** (*$f$-Divergence $D_f$* [20]). Let $f$ denote a convex function with $f(1) = 0$. $D_f(p, q) \equiv \int_{\mathcal{Y}} f(p(y), q(y))q(y) \, d[y]$, is a $f$-divergence function between two densities $p, q$.

$f$-divergence covers many well-known divergences. For example, $D_f$ reduces to KL divergence with $f(p, q) = (p/q) \log(p/q)$. We will assume that the function $f(p, q)$ in $D_f$ is differentiable w.r.t. $p$ and $q$.

We now analyze the distance between $\hat{\psi}_h$ and $\psi$ w.r.t. $D_f$. The following result provides an upper bound for $D_f$.

**Lemma 6** (**Upper bound of the divergence $D_f$**). *Suppose $D_f$ is a $f$-divergence such that $f(p, q) = 0$ if $p = q$. Then,*

$$D_f(\psi, \widehat{\psi}_h) \leq \int_{\mathcal{Y}} w(y) \left( \hat{\psi}_h(y) - \psi(y) \right) \, d[y],$$

*where $w(y) \equiv f_2'(\psi(y), \tilde{\psi}(y))\hat{\psi}_h(y)$, $f_2'(p, q) \equiv (\partial/\partial q)f(p, q)$, and $\tilde{\psi}_h(y) \equiv t\hat{\psi}_h(y) + (1-t)\psi(y)$ for some fixed $t \in [0, 1]$.*

By invoking Thm. 1, we derive an upper bound for $D_f(\psi, \widehat{\psi}_h)$ as follows:

**Theorem 2** (**Convergence rate of $\hat{\psi}_h$**). *Suppose the estimators for nuisances are consistent; i.e., $\|\nu - \hat{\nu}\| = o_P(1)$ for $\nu \in \eta = \{\pi, \xi, \theta\}$ for all $(w, z, x, y)$. Suppose $D_f$ is a $f$-divergence such that $f(p, q) = 0$ if $p = q$. Suppose $w(y)$ in Lemma 6 is finite. Then,*

$$D_f(\psi, \widehat{\psi}_h) \leq O_P \left( \sup_{y \in \mathcal{Y}} \left\{ R_2^k + B_y \right\} + 1/\sqrt{nh^d} + 1/\sqrt{n} \right), \tag{16}$$

*where $R_2^k$ is defined in Eq. (13) and $B_y$ is defined in Eq. (14).*

The following result asserts that the debiasedness property is exhibited w.r.t. $D_f$:

**Corollary 3** (**Debiasedness property of $\hat{\psi}_h$ w.r.t. $D_f$**). *Let $h = O(n^{-1/(d+4)})$. Suppose $D_f$ satisfies $f(p, q) = 0$ if $p = q$. Suppose $w(y)$ in Lemma 6 is finite. If nuisances $\{\hat{\pi}, \hat{\xi}, \hat{\theta}\}$ converges at $\{nh^d\}^{-1/4}$ rate for any $(w, z, x, y)$, then $D_f(\psi, \widehat{\psi}_h)$ converges to 0 at $\sqrt{nh^d}$-rate.*

## 4 Model-based approach

In this section, we develop a *model-based approach* for estimating the LTE density $\psi(y) = p(y_x|C)$. We will approximate $\psi$ with a class of distributions or a *density model* $\mathcal{G} = \{g(y; \beta) : \beta \in \mathbb{R}^b\}$ where $g(y; \beta) \in \mathcal{G}$ is differentiable w.r.t. $\beta$. Example density models include exponential family (e.g., Gaussian distribution), mixture of Gaussians, or more generally, mixture of exponential families. The choice of the density model may depend on domain knowledge. Alternatively, one may choose among a set of candidate density families using separate validation data or applying cross-validation. We adapt the model-based approach developed in [39] for estimating the causal density under the no unmeasured confounders assumption.

Given a density model $\mathcal{G}$, the best approximation for $\psi(y)$ is defined as $g(y; \beta_0) \in \mathcal{G}$ that achieves the minimum $f$-divergence to $\psi$:

$$\beta_0 \equiv \arg \min_{\beta \in \mathbb{R}^b} D_f(\psi(y), g(y; \beta)), \tag{17}$$

where $D_f$ is the $f$-divergence defined in Def. 2. Our goal is estimating $\beta_0$.

Consider $m(\beta; \psi) \equiv (\partial/\partial\beta)D_f(\psi(y), g(y;\beta))$. Definition of $\beta_0$ given in Eq. (17) implies that $m(\beta; \psi) = 0$ at $\beta = \beta_0$. We note that $m(\beta; \psi)$ serves as a *moment score function*. The closed-form expression of the score is given by [39]:

$$m(\beta; \psi) \equiv \int_{\mathcal{Y}} g'(y;\beta) \left\{ f_2'(\psi(y), g(y;\beta))g(y;\beta) + f(\psi(y), g(y;\beta)) \right\} \, d[y], \qquad (18)$$

where $g'(y;\beta) = (\partial/\partial\beta)g(y;\beta)$ and $f_2'(p, q) \equiv (\partial/\partial q)f(p, q)$.

To construct a DML estimator based on the score function $m(\beta; \psi)$, we first derive an influence function for the score:

**Lemma 7 (Influence Function for $m(\beta, \psi)$).** *An influence function for $m(\beta; \psi)$ in Eq. (18), denoted $\phi_m$, is given by*

$$\phi_m(\beta; \eta = \{\pi, \xi, \theta\}, \psi) \equiv \phi(\eta, \psi)[R_f(Y; \beta, \psi)], \qquad (19)$$

*where $\phi(\eta, \psi)[\cdot]$ is defined in Eq. (9), and*

$$R_f(Y; \beta, \psi) \equiv g'(Y;\beta) \left\{ f_{21}''(\psi(Y), g(Y;\beta))g(Y;\beta) + f_1'(\psi(Y), g(Y;\beta)) \right\},$$

*where $g'(y;\beta) \equiv (\partial/\partial\beta)g(y;\beta)$, $f_1'(p, q) \equiv (\partial/\partial p)f(p, q)$ and $f_{21}''(p, q) \equiv (\partial/\partial p)f_2'(p, q)$.*

We derive a Neyman orthogonal score based on the moment score $m(\beta, \psi)$ and its influence function $\phi_m(\beta, \eta, \psi)$:

**Lemma 8 (Neyman orthogonal score for $\beta$).** *A Neyman orthogonal score for estimating $\beta$, denoted $\varphi(\beta'; (\eta = \{\pi, \xi, \theta\}, \psi))$, is given by*

$$\varphi(\beta'; (\eta = \{\pi, \xi, \theta\}, \psi)) \equiv m(\beta', \psi) + \phi_m(\beta, \eta, \psi), \qquad (20)$$

*where $\phi_m(\beta, \eta, \psi)$ is defined in Eq. (19).*

Given the orthogonal score $\varphi(\beta'; (\eta, \psi))$ in Eq. (20), we propose the following estimator for $\beta$, named 'MLTE' (model-based estimator for LTE):

**Definition 3 (MLTE estimator for $\beta$).** *Let $\varphi(\beta'; \eta = \{\pi, \xi, \theta\}, \psi)$ be the Neyman orthogonal score for $\beta$ given in Eq. (20). Let $\{\mathcal{D}, \mathcal{D}'\}$ denote the randomly split halves of the samples, where $|\mathcal{D}| = |\mathcal{D}'| = n$. Let $\hat{\eta} = \{\hat{\pi}, \hat{\xi}, \hat{\theta}\}$ denote the estimators for the nuisance $\eta$ using $\mathcal{D}'$. Then, the MLTE estimator for $\beta$, denoted $\hat{\beta}$, is given as a solution satisfying $\mathbb{E}_{\mathcal{D}}\left[\varphi(\hat{\beta}; \hat{\eta}, \hat{\psi})\right] = o_P(n^{-1/2})$.*

To illustrate, we exemplify Eq. (18) and Lemma (7, 8) for the case where $D_f$ is a KL-divergence and $g(y; \beta = \{\mu, \sigma^2\})$ is a normal distribution. First, $m(\beta; \psi) = \{m_\mu(\mu; \psi), m_\sigma(\sigma^2; \psi, \mu)\}$, where $m_\mu(\mu; \psi, \sigma) = (1/\sigma^2)(\psi[Y] - \mu)$ and $m_\sigma(\sigma^2; \psi, \mu) = (0.5/\sigma^4)\left(\sigma^2 - \psi[(Y - \mu)^2]\right)$. We note that $\hat{\mu}_m \equiv \hat{\psi}[Y]$ and $\hat{\sigma}_m^2 \equiv \hat{\psi}[(Y - \hat{\mu})^2]$ are estimators for $\beta_0 = \{\mu_0, \sigma_0^2\}$ for the score $m(\beta; \psi)$.

Also, $R_f(Y; \beta, \psi) \equiv -(\partial/\partial\beta)\log(g(Y;\beta)) = \{R_f(Y; \mu, \psi), R_f(Y; \sigma^2, \psi)\}$, where $R_f(Y; \mu, \psi) \equiv (\mu - Y)/\sigma^2$ and $R_f(Y; \sigma^2, \psi) \equiv 0.5\{\sigma^2 - (Y - \mu)^2\}/\sigma^4$. Then, the Neyman orthogonal score is given as $\varphi(\mu; \sigma^2, \eta, \psi) = (1/\sigma^2)\{\mu - \psi[Y] - \phi(\eta, \psi)[Y]\}$ and $\varphi(\sigma^2; \mu, \eta, \psi) = (0.5/\sigma^4)\{\sigma^2 - \psi[(Y - \mu)^2] - \phi(\eta, \psi)[(Y - \mu)^2]\}$. Finally, solutions for $\varphi(\mu; \sigma^2, \eta, \psi)$ and $\varphi(\sigma^2, \mu; \eta, \psi)$ are given by $(\hat{\mu}, \hat{\sigma}^2)$, where, for $\phi[\cdot]$ in Eq. (9), $\hat{\mu} = \hat{\psi}[Y] + \mathbb{E}_{\mathcal{D}}\left[\phi(\hat{\eta}, \hat{\psi})[Y]\right]$ and $\hat{\sigma}^2 = \psi[(Y - \hat{\mu})^2] + \mathbb{E}_{\mathcal{D}}\left[\phi(\hat{\eta}, \hat{\psi})[(Y - \hat{\mu})^2]\right]$.

The MLTE estimator in Def. 3 is consistent provided that nuisances estimates $\hat{\eta}$ are consistent [14, Thm.4]. Such $\hat{\beta}$ is known to achieve debiasedness [13], since $\hat{\beta}$ is a DML estimator. Specifically,

**Theorem 3 (Convergence rate of $\hat{\beta}$).** *Let $\varphi(\beta'; (\eta = \{\pi, \xi, \theta\}, \psi)$ be given in Eq. (20). Let $\phi_m(\beta, \eta, \psi)$ be given in Eq. (19). Let $\beta_0, \eta_0, \psi_0$ denote the true parameters. Let $\hat{\beta}$ be the MLTE estimator for $\beta$ defined in Def. 3. Suppose (1) $R_f(y; \beta, \psi)$ is bounded and $R_f'(y; \beta, \psi) \equiv (\partial/\partial\psi)R_f(y; \beta, \psi) < \infty$; (2) There exists a function $H(y) < \infty$ s.t. $\sup_{\beta, \psi} \max\{R_f(y; \beta, \psi), R_f'(y; \beta, \psi)\} = O(H(y))$; (3) $\{\varphi(\beta; (\eta, \psi))\}$ is Donsker[4] w.r.t. $\beta$ for*

---

[4]A function class where complexities are restricted. See Def. S.1 in the Appendix for the definition. Donsker class include Sobolev, Bounded monotone, Lipschitz class, etc.

the fixed $\eta$; **(3)** *The estimators are consistent:* $\hat{\beta} - \beta_0 = o_P(1)$ *and* $\|\nu - \hat{\nu}\| = o_P(1)$ *for* $\nu \in \{\pi_z(w), \xi_x(z, w), \theta(x, z, w)[H(Y)]\}$ *for all* $(w, z, x, y)$; *and* **(4)** $\mathbb{E}_P[\varphi(\beta; (\eta, \psi))]$ *is differentiable w.r.t.* $\beta$ *at* $\beta = \beta_0$ *with non-singular matrix* $M(\beta_0, (\eta, \psi)) \equiv (\partial/\partial\beta)|_{\beta=\beta_0}\mathbb{E}_P[\varphi(\beta; (\eta, \psi))]$ *for all* $(\eta, \psi)$, *where* $M(\beta_0, (\hat{\eta}, \hat{\psi})) \xrightarrow{P} M \equiv M(\beta_0, (\eta_0, \psi_0))$. *Then,*

$$\widehat{\beta} - \beta_0 = -M^{-1}\mathbb{E}_{\mathcal{D}}[\phi_m(\beta_0; (\psi_0, \eta_0))] + o_P(n^{-1/2}) + O_P(R_2^m),$$

*where*

$$R_2^m = \sum_z \left( \|\hat{\pi}_z - \pi_z\| \left\{ \left\|\hat{\theta}_z - \theta_z\right\| + \left\|\hat{\xi}_z - \xi_z\right\| \right\} + \left\|\hat{\xi}_z - \xi_z\right\|^2 + \left\|\theta_z - \hat{\theta}_z\right\|^2 + \left\|\hat{\xi}_z - \xi_z\right\| \left\|\theta_z - \hat{\theta}_z\right\| \right),$$

*where* $\pi_z \equiv \pi_z(W)$, $\xi_z \equiv \xi_x(z, W)$, *and* $\theta_z \equiv \theta(x, z, W)[H(Y)]$.

**Corollary 4 (Debiasedness property for $\hat{\beta}$).** *If nuisances* $\{\hat{\pi}, \hat{\xi}, \hat{\theta}\}$ *converges at* $n^{-1/4}$ *rate, then the target estimator* $\hat{\beta}$ *converges to* $\beta_0$ *at* $\sqrt{n}$-*rate.*

For the above example where $D_f$ is the KL divergence and $g(y; \beta)$ is a normal distribution, $H(Y) = Y$ for $R_f(y; \mu, \psi)$, and $H(Y) = Y^2$ for $R_f(y; \sigma^2, \psi)$.

## 5 Empirical applications

In this section, we apply the proposed methods to synthetic and real datasets. For the kernel-smoothing based approach, we compare KLTE with a baseline plug-in estimator ('kernel-smoothing'), where estimates of nuisances $\hat{\eta} = \{\hat{\pi}, \hat{\xi}, \hat{\theta}\}$ are plugged in the estimand Eq. (2). We use the Gaussian kernel. The bandwidth is set to $h = 0.5n^{-1/5}$. In estimating the density, we choose 200 equi-spaced points $\{y_{(i)}\}_{i=1}^{200}$ in $\mathcal{Y}$ and evaluate both estimators at $K_{h, y_{(i)}}$ for $i = 1, \cdots, 200$. For the model-based approach, we compare MLTE (e.g., $\hat{\mu}, \hat{\sigma}^2$) with a moment-score-based estimator (called 'moment'), defined as $\hat{\beta}_m$ satisfying $m(\hat{\beta}_m; \hat{\psi}) = o_P(n^{-1/2})$ (e.g., $\{\hat{\mu}_m, \hat{\sigma}_m^2\}$). We use KL divergence for $D_f$ and the normal distribution for $g(y; \beta)$. For both approaches, nuisances are estimated through a gradient boosting model XGBoost [11], which is known to be flexible.

### 5.1 Synthetic dataset

We applied the proposed estimators to estimate the LTE $p(y_x|C)$ where the true densities are given as in the 4th plot in Fig. 2. As shown in the ground-truth in Fig. 3a, true densities $p(y_{x^0}|C), p(y_{x^1}|C)$ are given as a mixture of four Gaussians. Estimated densities for Moment and MLTE are given in Fig. (3b, 3c). We note that model-based approaches fail to capture important characteristics (such as the number of modes) of the true density ('ground-truth' in Fig. 3a) because the assumed density class is misspecified. The 'kernel-smoothing' (Fig. 3d) captures only one of the modes having the highest densities, and this leads to misinterpretation of the true densities. KLTE (Fig. 3e) is able to capture the number, location, and scales of modes correctly.

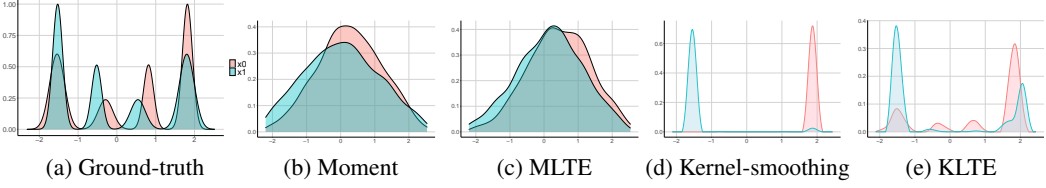

| (a) Ground-truth | (b) Moment | (c) MLTE | (d) Kernel-smoothing | (e) KLTE |

Figure 3: LTE estimation with a synthetic dataset. The ground-truth density is in (a). Red and Green for $x^0$ and $x^1$, respectively.

### 5.2 Application to 401(k) data

We applied the proposed estimators (KLTE and MLTE) on 401(k) data, where the data generating processes corroborate with Fig. 1. Monotonicity assumption holds naturally, since ineligible units ($Z = 0$) cannot participate ($X = 1$) in 401(k). In our analysis, we used the dataset introduced

by [2] containing 9275 individuals, which has been studied in [2, 17, 5, 47, 58, 64], to cite a few. Model-based approaches (Moment in Fig. 4a and MLTE in Fig. 4b) and kernel-smoothing based approaches (kernel-smoothing in Fig. 4c and KLTE in Fig. 4d) are implemented to analyze the data.

The model-based (Fig. (4a,4b)) and kernel-smoothing based (Fig. (4c,4d)) estimates both capture important characteristics of the distribution, such as mode, location, and scale parameters. The results of proposed estimators (MLTE and KLTE in Fig. (4b,4d)) are consistent with findings from previous analyses [2, 17, 5, 58]: The effects of the 401(k) participation (i.e., $X = 1$) on net financial assets are positive over the whole range of asset distributions. To connect to CDF method, we provide in Fig. 4e the CDF estimate induced by KLTE density estimation (Fig. 4a). We note that the CDF in Fig. 4e captures the nonconstant impact trend of the 401(k) participation on the net financial assets, which has been also described in the previous analyses [2, 17, 5, 58].

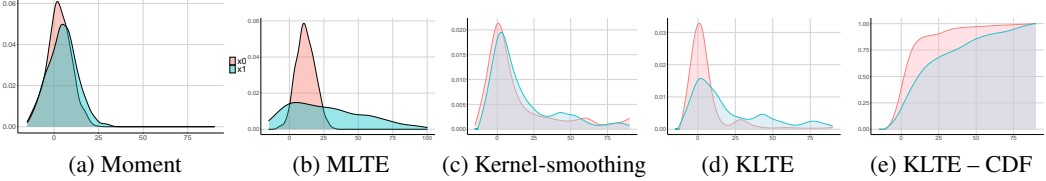

| (a) Moment | (b) MLTE | (c) Kernel-smoothing | (d) KLTE | (e) KLTE – CDF |

Figure 4: LTE of 401(k) participation ($X$) on net financial asset ($Y$). Red and Green for $x^0$ and $x^1$, respectively.

## 6   Conclusion

In this paper, we develop *kernel-smoothing-based* and *model-based* approaches for estimating the LTE density in the presence of instruments. For each approach, we give Neyman orthogonal scores (Lemma (2,8)) and constructed corresponding DML estimators (KLTE in Def. 1 and MLTE in Def. 3), that exhibit debiasedness property (Corol. (3, 4)). We demonstrated our work through synthetic and real datasets. The performance of model-based estimators depends critically on the choice of the density class. Kernel-based estimators do not have to make assumptions about the true density class but will suffer from the curse of dimensionality. This work is limited to settings where the monotonicity assumption holds, i.e., there are no defiers. One could perform sensitivity analyses on the impact of potential defiers to the estimates as conducted in [65, 36].

## Acknowledgements

We thank the reviewers for their feedback helping to improve this manuscript. Elias Bareinboim and Yonghan Jung were partially supported by grants from NSF IIS-1750807 (CAREER). Jin Tian was partially supported by ONR grant N000141712140.

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
