# Appendix – Double Machine Learning Density Estimation for Local Treatment Effects with Instruments

## A    IV Settings and LTE

In this work, we consider the IV setting represented by the causal graph $G$ in Fig. 1. It is common in the literature to define IV assumptions in terms of conditional independences among counterfactuals [2, 60, 47, 64], as given in the following:

**Assumption A.1** (**IV assumptions**)**.**

1. *Exclusion restriction*: $Y_{x,z} = Y_x$ almost surely for all $z, x$.

2. *Independence*: $Z \perp\!\!\!\perp (Y_x, X_z)|W$ for all $z, x$.

3. *Instruments relevance*: $P(X_{Z=1} = 1|W) \neq P(X_{Z=0} = 1|W)$ almost surely.

We show that the causal graph in Fig. 1 captures the set of IV assumptions in Assumption A.1.

**Lemma A.1.** *The causal graph $G$ in Fig. 1 satisfies the set of IV assumptions in Assumption A.1.*

*Proof.* We will show the first item. We have $Y_{x,z} = Y_{x,z,W_x} = Y_{x,W_x} = Y_x$, where the first equality is due to the composition property [50, Property 1 (pp. 229)], the second due to exclusion restrictions [50, Eq.(7.25)], and the third by composition.

We will show the second. We have $(Z_w \perp\!\!\!\perp \{W, X_{z,w}, Y_{x,w}\})$ by independence restrictions [50, Eq.(7.26)]. Then by the weak union graphoid axiom (Refer [50, pp.11]), $(Z_w \perp\!\!\!\perp \{X_{z,w}, Y_{x,w}\}|W)$, which leads to $(Z \perp\!\!\!\perp (Y_x, X_z)|W)$ by composition.

We will show the third. By $(Z \perp\!\!\!\perp X_z|W)$, $P(x_z|w) = P(x_z|w, z) = P(x|w, z)$, where the second equality is by composition. The third assumption is reflected by that $X$ is not independent of $Z$ given $W$ in $G$. $\square$

**Definition A.1** (**Local treatment effect (LTE) density**)**.** The *local treatment effect (LTE) density* is the density of outcome $Y$ under treatment $X = x$ among compliers (i.e., $X_{Z=1} = 1$ and $X_{Z=0} = 0$) denoted by $p(y_x|X_{Z=1} = 1, X_{Z=0} = 0)$. We will use $C = (X_{Z=1} = 1 \wedge X_{Z=0} = 0)$ to denote the event that a unit is a complier and write the LTE density as $p(y_x|C)$.

The LTE density $p(y_x|C)$ is known to be identifiable under monotonicity in the IV settings [31, 2]. In the notations of this paper, we present the identification results as follows, where for a given constant $a$ and a variable $X$, $x^a$ denotes the event $X = a$.

**Lemma A.2.** *In the causal graph $G$ in Fig. 1, $p(y_x|w, C)$ is identifiable under monotonicity and is given by*

$$p(y_x|w, C) = \frac{p(y|x, z^x, w)P(x|z^x, w) - p(y|w, x, z^{1-x})P(x|z^{1-x}, w)}{P(x^1|z^1, w) - P(x^1|z^0, w)}.$$

**Theorem A.1.** *In the causal graph $G$ in Fig. 1, the LTE density $p(y_x|C)$ is identifiable under monotonicity and is given by*

$$p(y_x|C) = \frac{\int_{\mathcal{W}}[p(y|x, z^x, w)P(x|z^x, w) - p(y|w, x, z^{1-x})P(x|z^{1-x}, w)]P(w) \, d[w]}{\int_{\mathcal{W}}[P(x^1|z^1, w) - P(x^1|z^0, w)]P(w) \, d[w]}$$

$$\equiv \frac{\mathbb{E}_P\left[p(y|x, z^x, W)P(x|z^x, W) - p(y|x, z^{1-x}, W)P(x|z^{1-x}, W)\right]}{\mathbb{E}_P\left[P(x^1|z^1, W) - P(x^1|z^0, W)\right]}.$$

## B    Proofs

**Notations**    We will use $P_\epsilon \equiv P(1 + \epsilon g)$, where $g$ is a mean zero bounded random function, to denote a parametric submodel for the probability measure $P$. Also, we note that the causal effect

$\psi[f(Y)]$ in Eq. (3) can be written as $\psi^{YX}[f(Y)]/\psi^X$, where $\psi^X$ and $\psi^{YX}[f(Y)]$ are defined in Eqs. (4,7).

We provide a formal definition of a function class called *Donsker class*, which is used throughout the proof.

**Definition S.1** (**Donsker Class** [63, page. 269]). *Let* $\mathbb{G}_n(f) \equiv \sqrt{n}(1/n)\sum_{i=1}^{n} f(\mathbf{v}_{(i)}) - \mathbb{E}_P[f(\mathbf{V})]$ *denote the empirical process evaluated at a measurable function* $f$. *A class of measurable functions* $\mathcal{F}$ *is called* (*P*-)*Donsker class if the sequence of processes* $\{\mathbb{G}_n(f); f \in \mathcal{F}\}$ *converges in distribution to a limit process* $\mathbb{G}$ *in the space* $\ell^\infty(\mathcal{F})$, *where* $\mathbb{G}$ *is the process such that, for all* $\epsilon > 0$, *there is a compact set* $S$ *such that* $P(\mathbb{G} \in S) > 1 - \epsilon$.

**Lemma S.1** ([63, Thm.5.31],[39, Lemma 3]). *Let* $\phi(\mathbf{V}; \theta, \eta)$ *denote a vector estimating function for target parameter* $\theta \in \mathbb{R}^p$ *and nuisance functions* $\eta \in H$ *for some function space* $H$. *Suppose* $\mathbb{E}_P[\phi(\mathbf{V}; \theta_0, \eta_0)] = 0$ *(where* $\theta_0, \eta_0$ *denote true parameters) and define the estimator* $\hat{\theta}$ *as a solution to* $\mathbb{E}_\mathcal{D}\left[\phi(\mathbf{V}; \hat{\theta}, \hat{\eta})\right] = o_P(n^{-1/2})$, *where* $\eta$ *is estimated on a separate independent sample. Assume*

1. $\{\phi(\mathbf{V}; \theta, \eta) : \theta \in \mathbb{R}^p\}$ *is Donsker for any fixed* $\eta$.

2. $\hat{\theta} - \theta_0 = o_P(1)$ *and* $\|\hat{\eta} - \eta\|_2 = o_P(1)$.

3. *The map* $\theta \mapsto \mathbb{E}_P[\phi(\mathbf{V}; \theta, \eta)]$ *is differentiable at* $\theta_0$ *uniformly in* $\eta$, *with non-singular matrix* $M(\theta_0, \eta) \equiv (\partial/\partial\theta)|_{\theta_0}\mathbb{E}_P[\phi(\mathbf{V}; \theta, \eta)]$, *where* $M(\theta_0, \hat{\eta}) \xrightarrow{P} M \equiv M(\theta_0, \eta_0)$.

*Then,*

$$\hat{\theta} - \theta_0 = -M^{-1}\mathbb{E}_\mathcal{D}[\phi(\mathbf{V}; \theta_0, \eta_0)] - M^{-1}\mathbb{E}_P[\phi(\mathbf{V}; \theta_0, \hat{\eta})] + o_P(n^{-1/2}).$$

## B.1  Proofs for Sec. 3

**Lemma S.2** ([28, Proof of Thm. 1]). *For a target estimand* $\gamma \equiv \mathbb{E}_P[\mathbb{E}_P[f(Y)|x^1, W] - \mathbb{E}_P[f(Y)|x^0, W]]$ *for binary* $X \in \{0, 1\}$ *and* $f(\cdot) < \infty$, *an influence function* $\phi_\gamma$ *is given by*

$$\phi_\gamma \equiv \frac{\mathbb{1}_{x^1}(X) - \mathbb{1}_{x^0}(X)}{P(X|W)}(f(Y) - \mathbb{E}_P[f(Y)|X, W]) + (\mathbb{E}_P[f(Y)|x^1, W] - \mathbb{E}_P[f(Y)|x^0, W]) - \gamma.$$

**Lemma S.3.** *An influence function for* $\psi[f(Y)]$ *for* $f(Y) < \infty$ *is given by the mapping function in Eq. (9), which is*

$$\phi(\eta = \{\pi, \xi, \theta\}, \psi)[f(Y)] \equiv \frac{1}{\psi^X}(\mathcal{V}_{YX}(\{\pi, \theta\})[f(Y)] - \psi[f(Y)]\mathcal{V}_X(\{\pi, \xi\})).$$

*Proof.* We note that the estimand is given as $\psi[f(Y)] = \psi^{YX}[f(Y)]/\psi^X$, where $\psi^X$ and $\psi^{YX}[f(Y)]$ are defined in Eqs. (4,7).

By Lemma S.2, influence functions corresponding to $\psi^X$ and $\psi^{YX}[f(Y)]$, denoted $\phi_X$ and $\phi_{YX}[f(Y)]$ respectively, are given as

$$\phi_X \equiv \frac{\mathbb{1}_{z^1}(Z) - \mathbb{1}_{z^0}(Z)}{\pi_Z(W)}(\mathbb{1}_{x^1}(X) - \xi_{x_1}(Z, W)) + (\xi_{x_1}(z^1, W) - \xi_{x_1}(z^0, W)) - \psi^X \tag{B.1}$$

$$\begin{aligned}\phi_{YX}[f(Y)] \equiv &\frac{\mathbb{1}_{z^x}(Z) - \mathbb{1}_{z^{1-x}}(Z)}{\pi_Z(W)}(f(Y)\mathbb{1}_x(X) - \theta(x, Z, W)[f(Y)])\\ &+ (\theta(x, z^x, W)[f(Y)] - \theta(x, z^{1-x}, W)[f(Y)]) - \psi^{YX}[f(Y)].\end{aligned} \tag{B.2}$$

Then, by applying the chain rule for the Gateaux derivative (since the influence function is a Gateaux derivative), an influence function for $\psi[f(Y)] = \psi^{YX}[f(Y)]/\psi^X$ is given as

$$\frac{1}{\psi^X}\left(\phi_{YX}[f(Y)] - \psi[f(Y)]\phi_X\right)$$

$$= \frac{1}{\psi^X}\left(\mathcal{V}_{YX}[f(Y)] - \psi^{YX}[f(Y)] - \psi[f(Y)]\left(\mathcal{V}_X - \psi^X\right)\right)$$

$$= \frac{1}{\psi^X}\left\{\mathcal{V}_{YX}[f(Y)] - \psi[f(Y)]\mathcal{V}_X\right\} - \psi[f(Y)] + \psi[f(Y)]$$

$$= \frac{1}{\psi^X}\left(\mathcal{V}_{YX}[f(Y)] - \psi[f(Y)]\mathcal{V}_X\right).$$

$\square$

**Lemma B.1** (Restated Lemma 1). *Let $m(\psi'; \psi_h)$ be the score defined in Eq. (6). Then, an influence function for $\mathbb{E}_P\left[m(\psi'; \psi_h)\right]$, denoted $\phi_m$, is given by*

$$\phi_m(\eta = \{\pi, \xi, \theta\}, \psi) \equiv \phi(\eta, \psi)[K_{h,y}(Y)] \tag{B.3}$$

*where $\phi$ is given as*

$$\phi(\eta = \{\pi, \xi, \theta\}, \psi)[f(Y)] \equiv \frac{1}{\psi^X}\left(\mathcal{V}_{YX}(\{\pi, \theta\})[f(Y)] - \psi[f(Y)]\mathcal{V}_X(\{\pi, \xi\})\right)$$

*Proof.* Let $\phi_X$ denote the influence function corresponding to $\psi^X$, given in Eq. (B.1). This implies that $\mathbb{E}_P\left[\mathcal{V}_X\right] = \psi^X$. Then, equipped with the true nuisance for $\mathcal{V}_X$,

$$\mathbb{E}_P\left[m(\psi'; \psi_h)\right] = \mathbb{E}_P\left[\frac{1}{\psi^X}\left(\psi_h - \psi'\right)\mathcal{V}_X\right] = \frac{1}{\psi^X}\left(\psi_h - \psi'\right)\mathbb{E}_P\left[\mathcal{V}_X\right] = \psi_h - \psi'.$$

Then, the influence function for $\mathbb{E}_P\left[m(\psi'; \psi_h)\right]$ coincides with the influence function for $\psi_h$, which is given by Eq. (B.3) based on Lemma S.3. $\square$

**Lemma B.2** (Restated Lemma 2). *Let $m(\psi'; \psi_h)$ be the score function in Eq. (6), and $\phi_m(\eta = \{\pi, \xi, \theta\}, \psi_h)$ be the influence function for $\mathbb{E}_P\left[m(\psi'; \psi_h)\right]$ given in Eq. (10). Then, a Neyman orthogonal score for $\psi_h$ is given as $\varphi(\psi'; \eta = \{\pi, \xi, \theta\}) \equiv m(\psi'; \psi_h) + \phi_m(\eta, \psi)$; Specifically,*

$$\varphi(\psi'; \eta = \{\pi, \xi, \theta\}) = \frac{1}{\psi^X}\left(\mathcal{V}_{YX}(\{\pi, \theta\})[K_{h,y}(Y)] - \psi'\mathcal{V}_X(\{\pi, \xi\})\right). \tag{B.4}$$

*Proof.* For a score function for $\psi$, denoted $m(\cdot)$, and the influence function of $\mathbb{E}_P\left[m(\cdot)\right]$, denoted $\phi_m(\cdot)$, a Neyman orthogonal score for $\psi$ is given as $m + \phi_m$ [14, Thm. 1]. Applying this, $m(\psi'; \psi_h) + \phi_m(\eta, \psi_h)$ is a Neyman orthogonal score. Specifically,

$$\varphi(\psi'; \eta = \{\pi, \xi, \theta\})$$
$$= m(\psi'; \psi_h) + \phi_m(\eta, \psi_h)$$
$$= \frac{1}{\psi^X}\left(\psi[K_{h,y}(Y)] - \psi'\right)\mathcal{V}_X + \frac{1}{\psi^X}\left(\mathcal{V}_{YX}(\{\pi, \theta\})[K_{h,y}(Y)] - \psi[K_{h,y}(Y)]\mathcal{V}_X(\{\pi, \xi\})\right)$$
$$= \frac{1}{\psi_X}\left(\mathcal{V}_{YX}(\eta = \{\pi, \theta\})[K_{h,y}(Y)] - \psi'\mathcal{V}_X(\{\pi, \xi\})\right).$$

$\square$

**Lemma B.3** (Restated Lemma 3). *For any fixed $y \in \mathcal{Y}$, suppose the estimators for nuisances are consistent; i.e., $\|\nu - \hat{\nu}\| = o_P(1)$ for $\nu \in \eta = \{\pi, \xi, \theta\}$ for all $(w, z, x)$. Suppose $h < \infty$, and $nh^d \to \infty$ as $n \to \infty$. Then,*

$$\hat{\psi}_h(y) - \psi_h(y) = O_P\left(1/\sqrt{nh^d} + R_2^k + 1/\sqrt{n}\right),$$

*where*

$$R_2^k \equiv \sum_z \|\hat{\pi}_z - \pi_z\|\left\{\left\|\hat{\theta}_z - \theta_z\right\| + \left\|\hat{\xi}_z - \xi_z\right\|\right\}, \tag{B.5}$$

*where $\pi_z \equiv \pi_z(W)$, $\xi_z \equiv \xi_x(z, W)$ and $\theta_z \equiv \theta(x, z, W)[K_{h,y}(Y)]$.*

*Proof.* We note that the condition $nh^d \to \infty$ means that $h = O(n^{-\alpha})$ for some $\alpha < 1/d$. $h < \infty$ implies that such $h$ is either constant or decreasing function over $n$. Combining, the condition implies $h = O(n^{-\alpha})$ for $\alpha \in [0, 1/d)$.

We recall that $\psi^X, \psi^{YX}$ are defined in Eq. (4.7) and $\mathcal{V}_X, \mathcal{V}_{YX}$ are defined in Eq. (5,8).

Now, we will prove this Lemma through the master result in Lemma S.1. The KLTE estimator $\hat{\psi}_h$ in Eq. (12) satisfies $\mathbb{E}_{\mathcal{D}}\left[\varphi(\hat{\psi}_h, \hat{\eta})\right] = o_P(n^{-1/2})$, because

$$\mathbb{E}_{\mathcal{D}}\left[\varphi(\hat{\psi}_h, \hat{\eta})\right] = \frac{1}{\psi^X(\hat{\xi})} \left(\mathbb{E}_{\mathcal{D}}\left[\mathcal{V}_{YX}(\{\hat{\pi}, \hat{\theta}\})[K_{h,y}(Y)]\right] - \hat{\psi}_h \mathbb{E}_{\mathcal{D}}\left[\mathcal{V}_X(\{\hat{\pi}, \hat{\xi}\})\right]\right)$$

$$= \frac{1}{\psi^X(\hat{\xi})} \left(\mathbb{E}_{\mathcal{D}}\left[\mathcal{V}_{YX}(\{\hat{\pi}, \hat{\theta}\})[K_{h,y}(Y)]\right] - \frac{\mathbb{E}_{\mathcal{D}}\left[\mathcal{V}_{YX}(\{\hat{\pi}, \hat{\theta}\})[K_{h,y}(Y)]\right]}{\mathbb{E}_{\mathcal{D}}\left[\mathcal{V}_X(\{\hat{\pi}, \hat{\xi}\})\right]} \mathbb{E}_{\mathcal{D}}\left[\mathcal{V}_X(\{\hat{\pi}, \hat{\xi}\})\right]\right)$$

$$= 0.$$

The Neyman orthogonal score function $\varphi$ in Lemma 2 satisfies the assumptions in Lemma S.1, since $\varphi$ is a linear function of $\psi$ when nuisances are fixed. Also, $M$ in Lemma S.1 is given as $-1$, which can be witnessed by the following:

$$M(\psi_0, \eta) = (\partial/\partial\psi')|_{\psi_0} \frac{1}{\psi^X} \mathbb{E}_P\left[\{\mathcal{V}_{YX} - \psi'\mathcal{V}_X\}\right] = -\frac{1}{\psi^X} \mathbb{E}_P\left[\mathcal{V}_X\right],$$

and, with the true nuisance, $M = M(\psi_0, \eta_0) = -1$ since $\mathbb{E}_P\left[\mathcal{V}_X\right] = \psi^X$.

Then, by the result of Lemma S.1,

$$\hat{\psi}_h - \psi_h = \mathbb{E}_{\mathcal{D}}\left[\phi_m(\psi_h, \eta)\right] + \mathbb{E}_P\left[\phi_m(\psi_h, \hat{\eta})\right] + o_P(n^{-1/2}).$$

We will first study the convergence behavior of $\mathbb{E}_{\mathcal{D}}\left[\phi_m(\psi_h, \eta)\right]$. We will show that $\mathbb{E}_P\left[\mathbb{E}_{\mathcal{D}}\left[\phi_m(\psi_h, \eta)\right]\right] = O\left(1/\sqrt{nh^d}\right)$. Then, the $\sqrt{nh^d}$-consistency of $\mathbb{E}_{\mathcal{D}}\left[\phi_m(\psi_h, \eta)\right]$ (i.e., $\mathbb{E}_{\mathcal{D}}\left[\phi_m(\psi_h, \eta)\right] = O_P(1/\sqrt{nh^d})$) can be shown immediately by the Markov inequality. This implies that $\mathbb{E}_{\mathcal{D}}\left[\phi_m(\psi_h, \eta)\right]$ converges if $nh^d \to \infty$.

Let $\phi_m(V_i, \psi, \eta)$ denote the influence function evaluated at $V_i \in \mathcal{D}$.

Consider the following:

$$\mathbb{E}_P\left[|\mathbb{E}_{\mathcal{D}}\left[\phi_m(\psi_h, \eta)\right]|\right] \leq \sqrt{\mathbb{E}_P\left[(\mathbb{E}_{\mathcal{D}}\left[\phi_m(\psi_h, \eta)\right])^2\right]}$$

$$= \sqrt{\text{var}_P\left(\mathbb{E}_{\mathcal{D}}\left[\phi_m(\psi_h, \eta)\right]\right)}$$

$$= \sqrt{(1/n)\mathbb{E}_P\left[\phi_m^2(\psi_h, \eta)\right]},$$

where the first inequality is by Cauchy-Schwarz inequality, the second and third equality are from the iid assumption and $\mathbb{E}_P\left[\phi_m\right] = 0$.

We note that

$$\phi_m = \frac{1}{\psi^X}\left(\mathcal{V}_{YX}[K_{h,y}(Y)] - \psi_h\mathcal{V}_X\right)$$

$$= \frac{1}{\psi^X}\left(\mathcal{V}_{YX}[K_{h,y}(Y)] - \psi_h\mathcal{V}_X\right) + \underbrace{\frac{\psi^{YX}[K_{h,y}(Y)]}{\psi^X} - \frac{\psi^X}{\psi^X}\psi_h}_{=0}$$

$$= \frac{1}{\psi^X}\left(\{\mathcal{V}_{YX}[K_{h,y}(Y)] - \psi^{YX}[K_{h,y}(Y)]\} - \psi_h\{\mathcal{V}_X - \psi^X\}\right)$$

$$= \frac{1}{\psi^X}\left(\phi_{YX}[K_{h,y}(Y)] - \psi_h\phi_X\right).$$

Next,

$$\mathbb{E}_P\left[\phi_m^2(\psi_h, \eta)\right] = \mathbb{E}_P\left[\frac{1}{\psi_X^2}\left\{\phi_{XY}[K_{h,y}(Y)] - \psi_h\phi_X\right\}^2\right]$$

$$= \frac{1}{\psi_X^2}\mathbb{E}_P\left[\left\{\phi_{XY}[K_{h,y}(Y)] - \psi_h\phi_X\right\}^2\right]$$

$$= \frac{1}{\psi_X^2}\mathbb{E}_P\left[\phi_{XY}^2[K_{h,y}(Y)] + \psi_h^2\phi_X^2 - 2\phi_{XY}[K_{h,y}(Y)]\phi_X\psi_h\right].$$

We first analyze $\mathbb{E}_P\left[\phi_{XY}^2[K_{h,y}(Y)]\right] = \mathrm{var}_P[\phi_{XY}[K_{h,y}(Y)]]$. By [28, Thm. 1],

$$\mathrm{var}_P[\phi_{XY}[K_{h,y}(Y)]] = \mathbb{E}_P\left[\frac{\mathrm{Var}_P\left(K_{h,y}(Y)\mathbb{1}_x(X)|z^x, W\right)}{\pi_{z^x}(W)} + \frac{\mathrm{Var}_P\left(K_{h,y}(Y)\mathbb{1}_x(X)|z^{1-x}, W\right)}{\pi_{z^{1-x}}(W)}\right]$$

$$+ \mathbb{E}_P\left[\left\{\mathbb{E}_P\left[K_{h,y}(Y)\mathbb{1}_x(X)|z^x, W\right] - \mathbb{E}_P\left[K_{h,y}(Y)\mathbb{1}_x(X)|z^{1-x}, W\right] - \psi^{YX}[K_{h,y}]\right\}^2\right].$$

First,

$$\mathbb{E}_P\left[\mathrm{Var}_P\left(K_{h,y}(Y)\mathbb{1}_x(X)|z^x, W\right)\right] \leq \mathrm{var}_P\left(K_{h,y}(Y)\mathbb{1}_x(X)|z^x\right)$$

$$\leq \mathbb{E}_P\left[K_{h,y}^2(Y)\mathbb{1}_x(X)|z^x\right]$$

$$\leq \mathbb{E}_P\left[K_{h,y}^2(Y)|x, z^x\right]$$

$$= \int_{\mathcal{Y}} K_{h,y}^2(y')p(y|x, z^x)\,d[y']$$

$$\leq \int_{\mathcal{Y}} K_{h,y}^2(y')\,d[y']$$

$$= \frac{1}{h^{2d}}\int_{\mathcal{Y}} K^2\left(\frac{y'-y}{h}\right)\,d[y']$$

$$= \frac{1}{h^d}\int_{\mathcal{Y}} K^2(u)\,d[u]$$

$$= O\left(1/h^d\right). \tag{B.6}$$

The 1st equality holds by Law of total variance, the 2nd and 3rd by the standard algebra, the 5th by the assumption that $p(y|x, z^x)$ is bounded, and the remaining parts from the change of a variable in the integral computation.

Also,

$$\mathbb{E}_P\left[\left\{\mathbb{E}_P\left[K_{h,y}(Y)\mathbb{1}_x(X)|z^x, W\right] - \mathbb{E}_P\left[K_{h,y}(Y)\mathbb{1}_x(X)|z^{1-x}, W\right] - \psi^{YX}[K_{h,y}]\right\}^2\right]$$

$$= \mathrm{var}_P\left(\left\{\mathbb{E}_P\left[K_{h,y}(Y)\mathbb{1}_x(X)|z^x, W\right] - \mathbb{E}_P\left[K_{h,y}(Y)\mathbb{1}_x(X)|z^{1-x}, W\right]\right\}\right)$$

$$\leq 2\sup_{z\in\{0,1\}}\mathrm{var}_P\left(\mathbb{E}_P\left[K_{h,y}(Y)\mathbb{1}_x(X)|z, W\right]\right)$$

$$= O(1/h^d),$$

where the first (in)equality is by the definition of the variance, the second by the linear combination of the variance, and the last by Eq. (B.6). Therefore, $\mathrm{var}_P[\phi_{XY}[K_{h,y}(Y)]] = O(1/h^d)$.

Next, we will study $\mathbb{E}_P\left[\psi_h^2\phi_X^2\right]$. We first note that $\mathbb{E}_P\left[\psi_h^2\phi_X^2\right] = \psi_h^2\mathbb{E}_P\left[\phi_X^2\right] = O(\psi_h^2)$. Therefore, it suffices to analyze $O\left(\psi_h^2\right)$.

$$
\begin{aligned}
\psi_h^2 &= \left(\int_{\mathcal{Y}} K_{h,y}(y')\psi(y')\,d[y']\right)^2 \\
&\leq \int_{\mathcal{Y}} K_{h,y}^2(y')\psi^2(y')\,d[y'] \\
&\leq \int_{\mathcal{Y}} K_{h,y}^2(y')\,d[y'] \\
&= \int_{\mathcal{Y}} \frac{1}{h^{2d}}K^2\left(\frac{y'-y}{h}\right)\,d[y'] \\
&= \int_{\mathcal{U}} \frac{1}{h^d}K^2(u)\,d[u] \\
&= O(1/h^d),
\end{aligned}
$$

where the 2nd line inequality by the Cauchy-Schwarz inequality, the 3rd by the assumption that $\psi(y)$ is bounded, the fifth by the change of variables.

Finally, consider the term $-2\mathbb{E}_P\left[\phi_{YX}[K_{h,y}(Y)]\cdot\phi_X\cdot\psi_h\right]$. Note, $\mathbb{E}_P\left[\phi_{YX}[K_{h,y}(Y)]\cdot\phi_X\cdot\psi_h\right] = \psi_h\cdot\mathbb{E}_P\left[\phi_{YX}[K_{h,y}(Y)]\cdot\phi_X\right]$. We first consider $\mathbb{E}_P\left[\phi_{YX}[K_{h,y}(Y)]\cdot\phi_X\right]$:

$$
\begin{aligned}
\mathbb{E}_P\left[\phi_{YX}[K_{h,y}(Y)]\cdot\phi_X\right] &= \mathbb{E}_P\left[\phi_{YX}[K_{h,y}(Y)]\cdot\phi_X\right] \\
&\leq \sqrt{\mathbb{E}_P\left[\phi_{YX}^2[K_{h,y}(Y)]\right]\cdot\mathbb{E}_P\left[\phi_X^2\right]} \\
&= O\left(\sqrt{\mathbb{E}_P\left[\phi_{YX}^2[K_{h,y}(Y)]\right]}\right) = O\left(h^{-d/2}\right),
\end{aligned}
$$

where the last equality holds by Eq. (B.6).

Next, consider $\psi_h$:

$$
\begin{aligned}
\psi_h &\equiv \int_{\mathcal{Y}} K_{h,y}(y')\psi(y')\,d[y'] \\
&= \int_{\mathcal{Y}} \frac{1}{h}K\left(\frac{(y'-y)}{h}\right)\psi(y')\,d[y'] \\
&= \int_{\mathcal{U}} K(u)\psi(hu+y)\,d[u] \\
&= \int_{\mathcal{U}} K(u)\left(\psi(y) + hu\psi^{(1)}(y) + h^2u^2\psi^{(2)}(y) + O(h^2u^2)\right)\,d[u] \\
&= C + O(h^2),
\end{aligned}
$$

for some constant $C$, The 4th line equality holds by the differentiability assumption of $\psi$, and the last equality holds since $\psi(y)$ is bounded and twice differentiable. Combining, we can rewrite the term $-2\mathbb{E}_P\left[\phi_{YX}[K_{h,y}(Y)]\cdot\phi_X\cdot\psi_h\right]$ as $O(h^{-d/2} + h^{-d/2}\cdot h^2)$.

Therefore,

$$
\mathbb{E}_P\left[\phi_m^2(\psi_h,\eta)\right] = O(h^{-d} + h^{-d/2} + h^{-d/2}h^2).
$$

With $h = O(n^{-\alpha})$ with $\alpha \in [0,1/d)$, we can rewrite

$$
\mathbb{E}_P\left[\phi_m^2(\psi_h,\eta)\right] = O(h^{-d} + h^{-d/2} + h^{-d/2}h^2) = O\left(n^{\alpha d}\right) = O(h^{-d}).
$$

This shows that

$$
\mathbb{E}_P\left[\mathbb{E}_{\mathcal{D}}\left[\phi_m(\psi_h,\eta)\right]\right] \leq \sqrt{(1/n)\mathbb{E}_P\left[\phi_m^2(\psi_h,\eta)\right]} = O\left(1/\sqrt{nh^d}\right).
$$

We now consider $\mathbb{E}_P\left[\phi_m(\psi_h, \hat{\eta})\right]$.

$$\mathbb{E}_P\left[\phi_m(\psi_h, \hat{\eta})\right]$$
$$= \mathbb{E}_P\left[\frac{1}{\hat{\psi}_X}\left(\hat{\mathcal{V}}_{YX}[K_{h,y}(Y)] - \psi[K_{h,y}(Y)]\hat{\mathcal{V}}_X\right)\right]$$
$$= \mathbb{E}_P\left[\frac{1}{\psi_X}\left(\hat{\mathcal{V}}_{YX}[K_{h,y}(Y)] - \psi[K_{h,y}(Y)]\hat{\mathcal{V}}_X\right) + \left(\frac{1}{\hat{\psi}_X} - \frac{1}{\psi_X}\right)\left(\hat{\mathcal{V}}_{YX}[K_{h,y}(Y)] - \psi[K_{h,y}(Y)]\hat{\mathcal{V}}_X\right)\right].$$
$$\text{(B.7)}$$

For further analysis, we consider $\mathbb{E}_P\left[\hat{\mathcal{V}}_{YX}[K_{h,y}(Y)] - \mathcal{V}_{YX}[K_{h,y}(Y)]\right]$. First, define

$$\mathcal{V}_{YX,(x,z)}(\pi, \theta)[f(Y)] \equiv \frac{\mathbb{1}_z(Z)}{\pi_Z(W)}\left(f(Y)\mathbb{1}_x(X) - \theta(x, Z, W)[f(Y)]\right) + \theta(x, z, W)[f(Y)].$$

Then, $\mathcal{V}_{YX}[f(Y)] = \mathcal{V}_{YX,(x,z^x)}[f(Y)] - \mathcal{V}_{YX,(x,z^{1-x})}[f(Y)]$. Now, consider $\mathbb{E}_P\left[\hat{\mathcal{V}}_{YX,(x,z)}[K_{h,y}(Y)] - \mathcal{V}_{YX,(x,z)}[K_{h,y}(Y)]\right]$. We have

$$\mathbb{E}_P\left[\mathcal{V}_{YX,(x,z)}(\hat{\pi}, \hat{\theta})[f(Y)] - \mathcal{V}_{YX,(x,z)}(\pi, \theta)[f(Y)]\right]$$
$$= \mathbb{E}_P\left[\frac{\mathbb{1}_z(Z)}{\hat{\pi}_Z(W)}\left(f(Y)\mathbb{1}_x(X) - \hat{\theta}(x, Z, W)[f(Y)]\right) + \hat{\theta}(x, z, W)[f(Y)] - \theta(x, z, W)[f(Y)]\right]$$
$$= \mathbb{E}_P\left[\frac{\mathbb{1}_z(Z)}{\hat{\pi}_Z(W)}\left(\theta(x, Z, W)[f(Y)] - \hat{\theta}(x, Z, W)[f(Y)]\right) + \left\{\hat{\theta}(x, z, W)[f(Y)] - \theta(x, z, W)[f(Y)]\right\}\right]$$
$$= \mathbb{E}_P\left[\frac{\pi_z(W)}{\hat{\pi}_z(W)}\left(\theta(x, z, W)[f(Y)] - \hat{\theta}(x, z, W)[f(Y)]\right) + \left\{\hat{\theta}(x, z, W)[f(Y)] - \theta(x, z, W)[f(Y)]\right\}\right]$$
$$= \mathbb{E}_P\left[\left(\theta(x, z, W)[f(Y)] - \hat{\theta}(x, z, W)[f(Y)]\right)\left(1 - \frac{\pi_z(W)}{\hat{\pi}_z(W)}\right)\right]$$
$$= \mathbb{E}_P\left[\left(\theta(x, z, W)[f(Y)] - \hat{\theta}(x, z, W)[f(Y)]\right)\left(\frac{\hat{\pi}_z(W) - \pi_z(W)}{\hat{\pi}_z(W)}\right)\right]$$
$$= O_P\left(\left\|\theta(x, z, W)[f(Y)] - \hat{\theta}(x, z, W)[f(Y)]\right\|\left\|\hat{\pi}_z(W) - \pi_z(W)\right\|\right),$$

where the first and the second are by the fact that $\mathbb{E}_P\left[f(Y)\mathbb{1}_x(X)|W, Z, X\right] = \theta(x, Z, W)[f(Y)]$, the third is by taking an expectation over $Z$ conditioned on $W$, the fourth and the fifth by rearrangement, and the sixth by Cauchy-Schwarz inequality and Positivity. Then,

$$R_{YX} \equiv \mathbb{E}_P\left[\mathcal{V}_{YX}(\hat{\pi}, \hat{\theta})[f(Y)] - \mathcal{V}_{YX}(\pi, \theta)[f(Y)]\right]$$
$$= \sum_{z \in \{0,1\}} O_P\left(\left\|\theta(x, z, W)[f(Y)] - \hat{\theta}(x, z, W)[f(Y)]\right\|\left\|\hat{\pi}_z(W) - \pi_z(W)\right\|\right).$$

Also, let

$$\mathcal{V}_{X,x}(\pi, \xi) \equiv \frac{\mathbb{1}_z(Z)}{\pi_Z(W)}\left(\mathbb{1}_x(X) - \xi_x(Z, W)\right) + \xi_x(z, W).$$

Then, with the similar proof as above, we have

$$\mathbb{E}_P\left[\mathcal{V}_{X,x}(\hat{\pi}, \hat{\xi}) - \mathcal{V}_{X,x}(\pi, \xi)\right] = O_P\left(\left\|\xi_x(z, W) - \hat{\xi}_x(z, W)\right\|\left\|\hat{\pi}_z(W) - \pi_z(W)\right\|\right),$$

and

$$\mathbb{E}_P\left[\mathcal{V}_X(\hat{\pi}, \hat{\xi}) - \mathcal{V}_X(\pi, \xi)\right] = \sum_{z \in \{0,1\}} O_P\left(\left\|\xi_x(z, W) - \hat{\xi}_x(z, W)\right\|\left\|\hat{\pi}_z(W) - \pi_z(W)\right\|\right).$$

Recall $R_{YX} = \mathbb{E}_P \left[ \hat{\mathcal{V}}_{YX} - \mathcal{V}_{YX} \right]$ and let $R_X \equiv \mathbb{E}_P \left[ \hat{\mathcal{V}}_X - \mathcal{V}_X \right]$. Then, continuing from Eq. (B.7),

$$
\begin{aligned}
\text{Eq. (B.7)} &= \mathbb{E}_P \left[ \frac{1}{\psi^X} \left( \psi_{YX} + R_{YX} - \frac{\psi_{YX}}{\psi_X} (\psi_X + R_X) \right) + \left( \frac{1}{\hat{\psi}_X} - \frac{1}{\psi_X} \right) \left( \psi_{YX} + R_{YX} - \frac{\psi_{YX}}{\psi_X} (\psi_X + R_X) \right) \right] \\
&= \mathbb{E}_P \left[ \frac{1}{\psi_X} (R_{YX} - \psi R_X) + \left( \frac{1}{\hat{\psi}_X} - \frac{1}{\psi_X} \right) (R_{YX} - \psi R_X) \right] \\
&= O_P(R_{YX} + R_X) \\
&= O_P(R_2^k),
\end{aligned}
$$

where

$$
R_2^k = \sum_{z \in \{0,1\}} O_P \left( \|\hat{\pi}_z - \pi_z\| \left\{ \left\| \hat{\theta}_z - \theta_z \right\| + \left\| \hat{\xi}_z - \xi_z \right\| \right\} \right).
$$

Note the first equality is by $\mathbb{E}_P \left[ \hat{\mathcal{V}}_{YX} \right] = R_{YX} + \mathbb{E}_P \left[ \mathcal{V}_{YX} \right]$ and $\mathbb{E}_P \left[ \hat{\mathcal{V}}_X \right] = R_X + \mathbb{E}_P \left[ \mathcal{V}_X \right]$, the second by rearrangement, the third by Positivity, the fourth by the definition of $R_{YX}$ and $R_X$.

Summing up, we have shown that $\mathbb{E}_P \left[ \phi_m(\psi_h, \eta) \right] = O(1/\sqrt{nh^d})$ and $\mathbb{E}_P \left[ \phi_m(\psi_h, \hat{\eta}) \right] = O_P \left( R_2^k \right)$.

$\square$

**Corollary 1** ((Restated Corol. 1)). *If all nuisances $\{\hat{\pi}, \hat{\xi}, \hat{\theta}\}$ for any given $(w, z, x, y)$ converge at rate $\{nh^d\}^{-1/4}$, then the target estimator $\hat{\psi}_h(y)$ achieves $\sqrt{nh^d}$-rate convergence to $\psi_h$.*

*Proof.* This result follows immediately from Lemma 3. $\square$

**Theorem B.1** (Restated Thm. 1). *For any fixed $y \in \mathcal{Y}$, suppose the estimators for nuisances are consistent; i.e., $\|\nu - \hat{\nu}\| = o_P(1)$ for $\nu \in \eta = \{\pi, \xi, \theta\}$ for all $(w, z, x)$. Suppose $h < \infty$, and $nh^d \to \infty$ as $n \to \infty$. Then*

$$
\hat{\psi}_h(y) - \psi(y) = O_P \left( 1/\sqrt{nh^d} + R_2^k + 1/\sqrt{n} \right) + B_y, \tag{B.8}
$$

*where $B_y$ is defined in Eq. (14), and $R_2^k$ is defined in Eq. (13).*

*Proof.* This result follows immediately from Lemmas 3 and 4. $\square$

**Lemma B.5** (Restated Lemma 5). *The bandwidth $h$ that minimizes the error in Eq. (15) is $h = O(n^{-1/(d+4)})$. This choice of $h$ satisfies the assumption in Lemma. 3 that $nh^d \to \infty$.*

*Proof.* We note that the error in Eq. (15) w.r.t. $h$ is $O_P(1/\sqrt{nh^d} + h^2)$. Since the function $1/\sqrt{nh^d} + h^2$ is convex w.r.t. $h$ and the global minimum is at $h = n^{-1/(d+4)}$, the optimal $h$ minimizing the error is $h = O(n^{-1/(d+4)})$. Then, $O(nh^d) = O(n^{4/(d+4)})$, implying that $nh^d \to \infty$. $\square$

**Corollary 2** (Restated Corol. 2). *Let $h = O(n^{-1/(d+4)})$. If nuisances $\{\hat{\pi}, \hat{\xi}, \hat{\theta}\}$ converge at $\{nh^d\}^{-1/4}$ rate for any $(w, z, x, y)$, then the target estimator $\hat{\psi}_h(y)$ achieves $\sqrt{nh^d}$-rate convergence to $\psi$.*

*Proof.* It suffices to show that $B_y$ converges at $\sqrt{nh^d}$-rate with the choice of $h$ as in Lemma 5, since the rest is guaranteed by Corol. 1. We first note that $B_y = O(h^2)$. Since $O(nh^d) = O(n^{4/(d+4)})$, we have $O(1/\sqrt{nh^d}) = O(n^{-2/(d+4)}) = O(h^2)$. $\square$

**Lemma B.6** (Restated Lemma 6). *Suppose $D_f$ is a $f$-divergence such that $f(p, q) = 0$ if $p = q$. Then,*

$$
D_f(\psi, \hat{\psi}_h) \leq \int_{\mathcal{Y}} w(y) \left( \hat{\psi}_h(y) - \psi(y) \right) d[y],
$$

*where $w(y) \equiv f_2'(\psi(y), \tilde{\psi}(y)) \hat{\psi}_h(y)$, $f_2'(p, q) \equiv (\partial/\partial q) f(p, q)$, and $\tilde{\psi}_h(y) \equiv t\hat{\psi}_h(y) + (1 - t)\psi(y)$ for some fixed $t \in [0, 1]$.*

*Proof.* For $f(p, q)$, by applying Taylor's expansion, we have

$$f(p, q) = f(p, p) + f_2'(p, \tilde{p})(q - p),$$

for some fixed $\tilde{p} \in [p, q]$. Applying this idea,

$$D_f(\psi, \widehat{\psi}_h) = \int_{\mathcal{Y}} f(\psi(y), \hat{\psi}_h(y)) \hat{\psi}_h(y) \, d[y]$$

$$= \int_{\mathcal{Y}} \left\{ \underbrace{f(\psi(y), \psi(y))}_{=0} + f_2'(\psi(y), \tilde{\psi}(y)) \left( \hat{\psi}_h(y) - \psi(y) \right) \right\} \hat{\psi}_h(y) \, d[y],$$

$$= \int_{\mathcal{Y}} w(y) \left( \hat{\psi}_h(y) - \psi(y) \right) \, d[y],$$

were the second equality holds by Taylor's expansion on $f$, and the third equality by the given assumption that $f(p, q) = 0$ whenever $p = q$.

$\square$

**Theorem B.2** (Restated Thm. 2). *Suppose the estimators for nuisances are consistent; i.e., $\|\nu - \hat{\nu}\| = o_P(1)$ for $\nu \in \eta = \{\pi, \xi, \theta\}$ for all $(w, z, x, y)$. Suppose $D_f$ is a $f$-divergence such that $f(p, q) = 0$ if $p = q$. Suppose $w(y)$ in Lemma 6 is finite. Then,*

$$D_f(\psi, \widehat{\psi}_h) \leq O_P \left( \sup_{y \in \mathcal{Y}} \left\{ R_2^k + B_y \right\} + 1/\sqrt{nh^d} + 1/\sqrt{n} \right), \tag{B.9}$$

*where $R_2^k$ is defined in Eq. (13) and $B_y$ is defined in Eq. (14).*

*Proof.* Under the given conditions, with Thm. 1,

$$D_f(\psi, \widehat{\psi}_h) \leq \int_{\mathcal{Y}} w(y) \left( \hat{\psi}_h(y) - \psi(y) \right) \, d[y]$$

$$= \int_{\mathcal{Y}} w(y) \left( O_P \left( 1/\sqrt{nh^d} + R_2^k + 1/\sqrt{n} \right) + B_y \right) \, d[y]$$

$$= O_P(1/\sqrt{nh^d} + 1/\sqrt{n}) + \int_{\mathcal{Y}} (w(y) O_P(R_2^k) + B_y) \, d[y]$$

$$= O_P(1/\sqrt{nh^d} + 1/\sqrt{n}) + O_P \left( \sup_{y \in \mathcal{Y}} \left\{ R_2^k + B_y \right\} \right).$$

$\square$

**Corollary 3** (Restated Corol. 3). *Let $h = O(n^{-1/(d+4)})$. Suppose $D_f$ satisfies $f(p, q) = 0$ if $p = q$. Suppose $w(y)$ in Lemma 6 is finite. If nuisances $\{\hat{\pi}, \hat{\xi}, \hat{\theta}\}$ converges at $\{nh^d\}^{-1/4}$ rate for any $(w, z, x, y)$, then $D_f(\psi, \widehat{\psi}_h)$ converges to 0 at $\sqrt{nh^d}$-rate.*

*Proof.* This result follows immediately from Thm. 2.

$\square$

## B.2 Proofs for Sec. 4

We will use $\psi_p$ to denote $\psi$ as a functional for $p$. Let $p_\epsilon$ denote a parametric submodel. We will use $S_\epsilon$ to denote a score function for $p_\epsilon$.

**Lemma B.7** (Restated Lemma 7). *An influence function for $m(\beta; \psi)$ in Eq. (18), denoted $\phi_m$, is given by*

$$\phi_m(\beta; \eta = \{\pi, \xi, \theta\}, \psi) \equiv \phi(\eta, \psi)[R_f(Y; \beta, \psi)], \tag{B.10}$$

*where $\phi(\eta, \psi)[\cdot]$ is defined in Eq. (9), and*

$$R_f(Y; \beta, \psi) \equiv g'(Y; \beta) \left\{ f_{21}''(\psi(Y), g(Y; \beta)) g(Y; \beta) + f_1'(\psi(Y), g(Y; \beta)) \right\},$$

*where $g'(y; \beta) \equiv (\partial/\partial\beta) g(y; \beta)$, $f_1'(p, q) \equiv (\partial/\partial p) f(p, q)$ and $f_{21}''(p, q) \equiv (\partial/\partial p) f_2'(p, q)$.*

*Proof.* Let $\psi_\epsilon$ denote the estimand $\psi$ written w.r.t. the parametric submodel $p_\epsilon = p(1 + \epsilon g)$ where $g$ is a bounded mean-zero random function. Let $S_\epsilon \equiv ((\partial/\partial\epsilon)|_{\epsilon=0} \log p_\epsilon$.

Let

$$\overline{m}(y; \beta, \psi) \equiv g'(y; \beta) \{f_2'(\psi(y), g(y; \beta))g(y; \beta) + f(\psi(y), g(y; \beta))\}. \tag{B.11}$$

Note $m(\beta, \psi) = \int_\mathcal{Y} \overline{m}(y; \beta, \psi) \, d[y]$. Also, we note that $(\partial/\partial\psi)\overline{m}(y; \beta, \psi) = R_f(y; \beta, \psi)$.

Also, recall that an influence function for $\psi[f(Y)]$ (for $f(Y) < \infty$) is given as $\phi(\eta, \psi)[f(Y)]$ in Lemma S.3. Then, by the definition of the influence function, $\psi[f(Y)]$ satisfies the following,

$$(\partial/\partial\epsilon)|_{\epsilon=0}\psi_\epsilon[f(Y)] = \mathbb{E}_P\left[\phi(\psi, \eta)[f(Y)] \cdot S_\epsilon\right].$$

Now, we will prove that $\phi_m(\beta; \eta = \{\pi, \xi, \theta\}, \psi) \equiv \phi(\eta, \psi)[R_f(Y; \beta, \psi)]$ is a functional satisfying

$$(\partial/\partial\epsilon)|_{\epsilon=0}m(\beta, \psi) = \mathbb{E}_P\left[\phi(\psi, \eta)[R_f(Y; \beta, \psi)] \cdot S_\epsilon\right],$$

then this equation implies that $\phi(\eta, \psi)[R_f(Y; \beta, \psi)]$ is an influence function for the score $m(\beta, \psi)$.

This can be shown as follows:

$$
\begin{aligned}
&(\partial/\partial\epsilon)|_{\epsilon=0}m(\beta, \psi) \\
&= (\partial/\partial\epsilon)|_{\epsilon=0} \int_\mathcal{Y} \overline{m}(y; \beta, \psi) \, d[y] \\
&= \int_\mathcal{Y} (\partial/\partial\epsilon)|_{\epsilon=0}\overline{m}(y; \beta, \psi) \, d[y] \\
&= \int_\mathcal{Y} (\partial/\partial\epsilon)|_{\epsilon=0}\psi_\epsilon(y)(\partial/\partial\psi'(y))|_{\psi'=\psi}\overline{m}(y; \beta, \psi_\epsilon) \, d[y] \\
&= (\partial/\partial\epsilon)|_{\epsilon=0} \int_\mathcal{Y} \psi_\epsilon(y)R_f(y; \beta, \psi) \, d[y] \\
&= (\partial/\partial\epsilon)|_{\epsilon=0}\psi_\epsilon[R_f(Y; \beta, \psi)] \\
&= \mathbb{E}_P\left[\phi(\psi, \eta)[R_f(Y; \beta, \psi)] \cdot S_\epsilon\right],
\end{aligned}
$$

where the first equality is by the definition of $\overline{m}$, the second by the exchange of derivation/integration, the third by the chain rule, the fourth by the fact that $(\partial/\partial\psi)\overline{m}(y; \beta, \psi) = R_f(y; \beta, \psi)$ and the exchange of derivation/integration, the fifth by the definition of $\psi[f(Y)]$ in Eq. (9), the sixth by the definition of the influence function (i.e., the influence function for $\psi[f(Y)]$ is a function $\phi[f(Y)]$ satisfying $(\partial/\partial\epsilon)|_{\epsilon=0}\psi_\epsilon[f(Y)] = \mathbb{E}_P\left[\phi[f(Y)] \cdot S_\epsilon\right]$.

$\square$

**Lemma B.8** ((Restated Lemma 8)). *A Neyman orthogonal score for estimating $\beta$, denoted $\varphi(\beta'; (\eta = \{\pi, \xi, \theta\}, \psi))$, is given by*

$$\varphi(\beta'; (\eta = \{\pi, \xi, \theta\}, \psi)) \equiv m(\beta', \psi) + \phi_m(\beta, \eta, \psi), \tag{B.12}$$

*where $\phi_m(\beta, \eta, \psi)$ is defined in Eq. (19).*

*Proof.* We first note that $\mathbb{E}_P\left[m(\beta', \psi)\right] = m(\beta', \psi)$, because this is not a random function. Then, the influence function for $\mathbb{E}_P\left[m(\beta', \psi)\right]$ is given by Lemma 7. For any score function which expectation is zero at the true parameter, its addition with the influence function is a Neyman orthogonal score [14, Thm.1]. That is, $m(\beta', \psi) + \phi_m(\beta, \eta, \psi)$ is a Neyman orthogonal score. $\square$

**Theorem B.3** ((Restated Thm. 3)). *Let $\varphi(\beta'; (\eta = \{\pi, \xi, \theta\}, \psi)$ be given in Eq. (20). Let $\phi_m(\beta, \eta, \psi)$ be given in Eq. (19). Let $\beta_0, \eta_0, \psi_0$ denote the true parameters. Let $\hat\beta$ be the MLTE estimator for $\beta$ defined in Def. 3. Suppose (1) $R_f(y; \beta, \psi)$ is bounded and $R_f'(y; \beta, \psi) \equiv (\partial/\partial\psi)R_f(y; \beta, \psi) < \infty$; (2) There exists a function $H(y) < \infty$ s.t. $\sup_{\beta, \psi} \max\{R_f(y; \beta, \psi), R_f'(y; \beta, \psi)\} = O(H(y))$; (3) $\{\varphi(\beta; (\eta, \psi))\}$ is Donsker w.r.t. $\beta$ for the fixed $\eta$; (3) The estimators are consistent: $\hat\beta - \beta_0 = o_P(1)$ and $\|\nu - \hat\nu\| = o_P(1)$ for $\nu \in \{\pi_z(w), \xi_x(z, w), \theta(x, z, w)[H(Y)]\}$ for all $(w, z, x, y)$; and (4)*

$\mathbb{E}_P\left[\varphi(\beta;(\eta,\psi))\right]$ *is differentiable w.r.t.* $\beta$ *at* $\beta = \beta_0$ *with non-singular matrix* $M(\beta_0,(\eta,\psi)) \equiv (\partial/\partial\beta)|_{\beta=\beta_0}\mathbb{E}_P\left[\varphi(\beta;(\eta,\psi))\right]$ *for all* $(\eta,\psi)$, *where* $M(\beta_0,(\hat{\eta},\hat{\psi})) \xrightarrow{P} M \equiv M(\beta_0,(\eta_0,\psi_0))$. *Then,*

$$\widehat{\beta} - \beta_0 = -M^{-1}\mathbb{E}_{\mathcal{D}}\left[\phi_m(\beta_0;(\psi_0,\eta_0))\right] + o_P(n^{-1/2}) + O_P(R_2^m),$$

*where*

$$R_2^m = \sum_z \left(\|\hat{\pi}_z - \pi_z\|\left\{\left\|\hat{\theta}_z - \theta_z\right\| + \left\|\hat{\xi}_z - \xi_z\right\|\right\} + \left\|\hat{\xi}_z - \xi_z\right\|^2 + \left\|\theta_z - \hat{\theta}_z\right\|^2 + \left\|\hat{\xi}_z - \xi_z\right\|\left\|\theta_z - \hat{\theta}_z\right\|\right),$$

*where* $\pi_z \equiv \pi_z(W)$, $\xi_z \equiv \xi_x(z,W)$, *and* $\theta_z \equiv \theta(x,z,W)[H(Y)]$.

*Proof.* We follow the proof strategy used in [39, Lemma 1, Thm.3]. First,

$$\widehat{\beta} - \beta_0 = -M^{-1}\mathbb{E}_{\mathcal{D}}\left[\varphi(\beta_0,(\psi_0,\eta_0))\right] - M^{-1}\mathbb{E}_P\left[\varphi(\beta_0,(\widehat{\psi},\widehat{\eta}))\right] + o_P(n^{-1/2})$$

$$= -M^{-1}\mathbb{E}_{\mathcal{D}}\left[\phi_m(\beta_0,\{\psi_0,\eta_0\})\right] - M^{-1}\mathbb{E}_P\left[\varphi(\beta_0,(\widehat{\psi},\widehat{\eta}))\right] + o_P(n^{-1/2}), \quad \text{(B.13)}$$

where the first equality holds by Lemma S.1, and the second holds since $m(\beta_0,\psi_0) = 0$ by the moment condition in Eq. (18). Since $\mathbb{E}_{\mathcal{D}}\left[\phi_m(\beta_0,\eta_0,\psi_0)\right]$ converges to $N(0,\text{var}(\phi_m^2))$ in distribution at $\sqrt{n}$-rate, the only remaining term to analyze is

$$\mathbb{E}_P\left[\varphi(\beta_0,(\widehat{\psi},\widehat{\eta}))\right] = m(\beta_0,\hat{\psi}) + \mathbb{E}_P\left[\phi(\beta_0,(\widehat{\psi},\widehat{\eta}))[R_f(Y;\beta_0,\hat{\psi})]\right], \quad \text{(B.14)}$$

which can be analyzed as

$$\mathbb{E}_P\left[\phi(\beta_0,(\widehat{\psi},\widehat{\eta}))[R_f(Y;\beta_0)]\right]$$

$$= \mathbb{E}_P\left[\frac{1}{\hat{\psi}_X}\left\{\hat{\mathcal{V}}_{YX}[R_f(Y;\beta_0,\hat{\psi})] - \hat{\psi}[R_f(Y;\beta_0,\hat{\psi})]\hat{\mathcal{V}}_X\right\}\right]$$

$$= \mathbb{E}_P\left[\frac{1}{\hat{\psi}_X}\hat{\mathcal{V}}_{YX}[R_f(Y;\beta_0,\hat{\psi})]\right] - \mathbb{E}_P\left[\frac{1}{\hat{\psi}_X}\hat{\psi}[R_f(Y;\beta_0,\hat{\psi})]\hat{\mathcal{V}}_X\right]$$

$$= \mathbb{E}_P\left[\frac{1}{\hat{\psi}_X}\left\{\frac{\hat{\pi}_{z^x}(W)}{\pi_{z^x}(W)}\left\{\theta(x,z^x,W)[R_f(Y;\beta_0,\hat{\psi})] - \hat{\theta}(x,z^x,W)[R_f(Y;\beta_0,\hat{\psi})]\right\} + \hat{\theta}(x,z^x,W)R_f(Y;\beta_0,\hat{\psi})\right\}\right]$$
$$\text{(B.15)}$$

$$- \mathbb{E}_P\left[\frac{1}{\hat{\psi}_X}\left\{\frac{\hat{\pi}_{z^{1-x}}(W)}{\pi_{z^{1-x}}(W)}\left\{\theta(x,z^{1-x},W)[R_f(Y;\beta_0,\hat{\psi})] - \hat{\theta}(x,z^{1-x},W)[R_f(Y;\beta_0,\hat{\psi})]\right\} + \hat{\theta}(x,z^{1-x},W)[R_f(Y;\beta_0,\hat{\psi})]\right\}\right]$$
$$\text{(B.16)}$$

$$- \mathbb{E}_P\left[\frac{1}{\hat{\psi}_X}\hat{\psi}[R_f(Y;\beta_0,\hat{\psi})]\left\{\frac{\pi_{z^x}(W)}{\hat{\pi}_{z^x}(W)}\left\{\xi_x(z^x,W) - \hat{\xi}_x(z^x,W)\right\} + \hat{\xi}_x(z^x,W)\right\}\right] \quad \text{(B.17)}$$

$$+ \mathbb{E}_P\left[\frac{1}{\hat{\psi}_X}\hat{\psi}[R_f(Y;\beta_0,\hat{\psi})]\left\{\frac{\pi_{z^{1-x}}(W)}{\hat{\pi}_{z^{1-x}}(W)}\left\{\xi_x(z^{1-x},W) - \hat{\xi}_x(z^{1-x},W)\right\} + \hat{\xi}_x(z^{1-x},W)\right\}\right], \quad \text{(B.18)}$$

where

$$\text{(B.15)} = \mathbb{E}_P\left[\frac{1}{\hat{\psi}_X}\cdot\left\{\left(\frac{\hat{\pi}_{z^x}(W)}{\pi_{z^x}(W)} - 1\right)\left\{\theta(x,z^x,W)[R_f(Y;\beta_0,\hat{\psi})] - \hat{\theta}(x,z^x,W)[R_f(Y;\beta_0,\hat{\psi})]\right\}\right\}\right]$$
$$\text{(B.19)}$$

$$+ \mathbb{E}_P\left[\frac{1}{\hat{\psi}_X}\theta(x,z^x,W)[R_f(Y;\beta_0,\hat{\psi})]\right] \quad \text{(B.20)}$$

$$\text{(B.16)} = -\mathbb{E}_P\left[\frac{1}{\hat{\psi}_X}\left\{\left(\frac{\hat{\pi}_{z^{1-x}}(W)}{\pi_{z^{1-x}}(W)} - 1\right)\left\{\theta(x,z^{1-x},W)[R_f(Y;\beta_0,\hat{\psi})] - \hat{\theta}(x,z^{1-x},W)[R_f(Y;\beta_0,\hat{\psi})]\right\}\right\}\right]$$
$$\text{(B.21)}$$

$$- \mathbb{E}_P\left[\frac{1}{\hat{\psi}_X}\theta(x,z^{1-x},W)[R_f(Y;\beta_0,\hat{\psi})]\right] \quad \text{(B.22)}$$

$$(\text{B}.17) = -\mathbb{E}_P\left[\frac{1}{\hat{\psi}_X}\hat{\psi}[R_f(Y;\beta_0,\hat{\psi})]\left\{\left(\frac{\pi_{z^x}(W)}{\hat{\pi}_{z^x}(W)}-1\right)\left\{\xi_x(z^x,W)-\hat{\xi}_x(z^x,W)\right\}\right\}\right] \quad (\text{B}.23)$$

$$-\mathbb{E}_P\left[\frac{1}{\hat{\psi}_X}\hat{\psi}[R_f(Y;\beta_0,\hat{\psi})]\xi_x(z^x,W)\right] \quad (\text{B}.24)$$

$$(\text{B}.18) = \mathbb{E}_P\left[\frac{1}{\hat{\psi}_X}\hat{\psi}[R_f(Y;\beta_0,\hat{\psi})]\left\{\left(\frac{\pi_{z^{1-x}}(W)}{\hat{\pi}_{z^{1-x}}(W)}-1\right)\left\{\xi_x(z^{1-x},W)-\hat{\xi}_x(z^{1-x},W)\right\}\right\}\right]$$
$$(\text{B}.25)$$

$$+\mathbb{E}_P\left[\frac{1}{\hat{\psi}_X}\hat{\psi}[R_f(Y;\beta_0,\hat{\psi})]\xi_x(z^{1-x},W)\right] \quad (\text{B}.26)$$

First, consider the summation of (B.20,B.22,B.24,B.26):

Eq. (B.20) + Eq. (B.22) + Eq. (B.24) + Eq. (B.26)

$$= \mathbb{E}_P\left[\frac{1}{\hat{\psi}_X}\left\{\theta(x,z^x,W)[R_f(Y;\beta_0,\hat{\psi})]-\theta(x,z^{1-x},W)[R_f(Y;\beta_0,\hat{\psi})]\right\}\right]$$

$$-\mathbb{E}_P\left[\frac{1}{\hat{\psi}_X}\hat{\psi}[R_f(Y;\beta_0,\hat{\psi})]\left\{\xi_x(z^x,W)-\xi_x(z^{1-x},W)\right\}\right]$$

$$= \mathbb{E}_P\left[\frac{1}{\hat{\psi}_X}\left(\psi_{YX}[R_f(Y;\beta_0,\hat{\psi})]-\hat{\psi}[R_f(Y;\beta_0,\hat{\psi})]\cdot\psi_X\right)\right]$$

$$= \mathbb{E}_P\left[\frac{1}{\hat{\psi}_X}\left(\psi_{YX}[R_f(Y;\beta_0,\hat{\psi})]-\frac{\hat{\psi}_{YX}[[R_f(Y;\beta_0,\hat{\psi})]]}{\hat{\psi}_X}\cdot\psi_X\right)\right]$$

$$= \mathbb{E}_P\left[\frac{\psi_X}{\hat{\psi}_X}\left(\frac{\psi_{YX}[R_f(Y;\beta_0,\hat{\psi})]}{\psi_X}-\frac{\hat{\psi}_{YX}[R_f(Y;\beta_0,\hat{\psi})]}{\hat{\psi}_X}\right)\right]$$

$$= \mathbb{E}_P\left[\frac{\psi_X}{\hat{\psi}_X}\left(\psi[R_f(Y;\beta_0,\hat{\psi})]-\hat{\psi}[R_f(Y;\beta_0,\hat{\psi})]\right)\right].$$

$$= \mathbb{E}_P\left[\left\{\frac{\psi_X}{\hat{\psi}_X}-1\right\}\left(\psi[R_f(Y;\beta_0,\hat{\psi})]-\hat{\psi}[R_f(Y;\beta_0,\hat{\psi})]\right)\right]+\mathbb{E}_P\left[\left(\psi[R_f(Y;\beta_0,\hat{\psi})]-\hat{\psi}[R_f(Y;\beta_0,\hat{\psi})]\right)\right].$$
$$(\text{B}.27)$$

Then,

Eq. (B.14) $= m(\beta_0,\hat{\psi}) + $ Sum of (B.20, B.22, B.24, B.26) + Sum of (B.19, B.21, B.23, B.25)

$$= m(\beta_0,\hat{\psi}) + \mathbb{E}_P\left[\left(\psi[R_f(Y;\beta_0,\hat{\psi})]-\hat{\psi}[R_f(Y;\beta_0,\hat{\psi})]\right)\right] \quad (\text{B}.28)$$

$$+ \mathbb{E}_P\left[\left\{\frac{\psi_X}{\hat{\psi}_X}-1\right\}\left(\psi[R_f(Y;\beta_0,\hat{\psi})]-\hat{\psi}[R_f(Y;\beta_0,\hat{\psi})]\right)\right] + \text{Sum of (B.19,B.21,B.23,B.25).}$$
$$(\text{B}.29)$$

To analyze (B.28), we recall that $(\partial/\partial\psi)m(\beta_0,\psi) = \int_{\mathcal{Y}} R_f(y;\beta_0,\psi)\,d[y]$ and $m(\beta_0,\psi)=0$. Also, by Taylor's expansion to $\overline{m}(y;\beta,\psi)$ defined in Eq. (B.11),

$$\overline{m}(y;\beta_0,\psi) = \overline{m}(y;\beta_0,\hat{\psi}) + R_f(y;\beta,\hat{\psi})(\psi(y)-\hat{\psi}(y)) + R_f^{(1)}(y;\beta,\tilde{\psi})(\psi(y)-\hat{\psi}(y))^2,$$

where $R_f^{(1)}$ is a first derivative of $R_f$ w.r.t. $\psi$. This implies that

$$0 = m(\beta_0,\psi) = m(\beta_0,\widehat{\psi}) + \int_{\mathcal{Y}} R_f(y;\beta,\hat{\psi})\left(\psi(y)-\hat{\psi}(y)\right)\,d[y] + \int_{\mathcal{Y}} R_f^{(1)}(y;\beta,\tilde{\psi})\left(\psi(y)-\hat{\psi}(y)\right)^2\,d[y],$$

where $\tilde{\psi}$ is some unknown estimand within the interval $[\psi,\hat{\psi}]$. We obtain

$$-\int_{\mathcal{Y}} R_f^{(1)}(y;\beta,\tilde{\psi})\left(\psi(y)-\hat{\psi}(y)\right)^2\,d[y] = m(\beta_0,\widehat{\psi}) + \int_{\mathcal{Y}} R_f(y;\beta,\hat{\psi})\left(\psi(y)-\hat{\psi}(y)\right)\,d[y].$$

By taking expectations for both sides,

$$-\mathbb{E}_P\left[\int_{\mathcal{Y}} R_f^{(1)}(y;\beta,\tilde{\psi})\left(\psi(y)-\hat{\psi}(y)\right)^2 d[y]\right] = m(\beta_0,\widehat{\psi}) + \mathbb{E}_P\left[\int_{\mathcal{Y}} R_f(y;\beta,\hat{\psi})\left(\psi(y)-\hat{\psi}(y)\right) d[y]\right].$$

$$\text{(B.30)}$$

We have

$$-\int_{\mathcal{Y}} R_f^{(1)}(y;\beta,\tilde{\psi})\left(\psi(y)-\hat{\psi}(y)\right)^2 d[y] = O\left(\int_{\mathcal{Y}} R_f^{(1)}(y;\beta,\tilde{\psi})\left(\psi(y)-\hat{\psi}(y)\right)^2 d[y]\right)$$

$$= O\left(\int_{\mathcal{Y}} H(y)\left(\psi(y)-\hat{\psi}(y)\right)^2 d[y]\right)$$

$$= O\left(\int_{\mathcal{Y}} H^2(y)\left(\psi(y)-\hat{\psi}(y)\right)^2 d[y]\right)$$

$$= O\left(\left\|\psi[H(Y)]-\hat{\psi}[H(Y)]\right\|^2\right),$$

where the second equality is by the definition of $H(y)$, the third by $H(y)<\infty$, and the fourth by the definition of $L_2$ norm.

This implies that

$$\text{(B.28)} = -\mathbb{E}_P\left[\int_{\mathcal{Y}} R_f^{(1)}(y;\beta,\tilde{\psi})\left(\psi-\hat{\psi}\right)^2 d[y]\right] = O\left(\left\|\psi[H(Y)]-\hat{\psi}[H(Y)]\right\|^2\right),$$

where the first equality is by Eq. (B.30) and the second equality is by the above.

Also, Sum of (B.19,B.21,B.23,B.25) in (B.29) can be written as follows:

Sum of (B.19,B.21,B.23,B.25)

$$= \sum_{z\in\{0,1\}} O_P\left(\|\hat{\pi}_z(W)-\pi_z(W)\|\left\{\left\|\hat{\theta}(x,z,W)[R_f(Y;\beta_0,\hat{\psi})]-\theta(x,z,W)[R_f(Y;\beta_0,\hat{\psi})]\right\|+\left\|\hat{\xi}_x(z,W)-\xi_x(z,W)\right\|\right\}\right)$$

$$= \sum_{z\in\{0,1\}} O_P\left(\|\hat{\pi}_z(W)-\pi_z(W)\|\left\{\left\|\hat{\theta}(x,z,W)[H(Y)]-\theta(x,z,W)[H(Y)]\right\|+\left\|\hat{\xi}_x(z,W)-\xi_x(z,W)\right\|\right\}\right).$$

For simplicity, we assume, for any $x,z$,

$$O_P\left(\left\{\xi_x(z,W)-\hat{\xi}_x(z,W)\right\}\cdot\left\{\xi_x(1-z,W)-\hat{\xi}_x(1-z,W)\right\}\right) = \sum_{z\in\{0,1\}} O_P\left(\left\|\xi_x(z,W)-\hat{\xi}_x(z,W)\right\|^2\right), \text{ and}$$

$$O_P\left(\left\|\xi_x(z,W)-\hat{\xi}_x(z,W)\right\|\left\|\theta(x,1-z,W)[H(Y)]-\hat{\theta}(x,1-z,W)[H(Y)]\right\|\right)$$

$$= \sum_{z\in\{0,1\}} O_P\left(\left\|\xi_x(z,W)-\hat{\xi}_x(z,W)\right\|\left\|\theta(x,z,W)[H(Y)]-\hat{\theta}(x,z,W)[H(Y)]\right\|\right).$$

The other part of Eq. (B.29) is given as

$$\mathbb{E}_P\left[\left\{\frac{\psi_X}{\hat{\psi}_X} - 1\right\}\left(\psi[R_f(Y;\beta_0,\hat{\psi})] - \hat{\psi}[R_f(Y;\beta_0,\hat{\psi})]\right)\right]$$

$$= O_P\left(\left\|\psi^X - \hat{\psi}^X\right\|\left\|\psi[R_f(Y;\beta_0,\hat{\psi})] - \hat{\psi}[R_f(Y;\beta_0,\hat{\psi})]\right\|\right)$$

$$= O_P\left(\left\|\psi^X - \hat{\psi}^X\right\|\left\|\frac{\psi^{YX}[R_f(Y;\beta_0,\hat{\psi})]}{\psi^X} - \frac{\hat{\psi}^{YX}[R_f(Y;\beta_0,\hat{\psi})]}{\psi^X} + \frac{\hat{\psi}^{YX}[R_f(Y;\beta_0,\hat{\psi})]}{\psi^X} - \frac{\hat{\psi}^{YX}[R_f(Y;\beta_0,\hat{\psi})]}{\hat{\psi}^X}\right\|\right)$$

$$= O_P\left(\left\|\psi^X - \hat{\psi}^X\right\|\left(\left\|\psi^{YX}[R_f(Y;\beta_0,\hat{\psi})] - \hat{\psi}^{YX}[R_f(Y;\beta_0,\hat{\psi})]\right\| + \left\|\frac{1}{\psi^X} - \frac{1}{\hat{\psi}^X}\right\|\right)\right)$$

$$= O_P\left(\left\|\psi^X - \hat{\psi}^X\right\|\left(\left\|\psi^{YX}[R_f(Y;\beta_0,\hat{\psi})] - \hat{\psi}^{YX}[R_f(Y;\beta_0,\hat{\psi})]\right\| + \left\|\psi^X - \hat{\psi}^X\right\|\right)\right)$$

$$= O_P\left(\left\|\psi^X - \hat{\psi}^X\right\|^2\right) + O_P\left(\left\|\psi^X - \hat{\psi}^X\right\|\left\|\psi^{YX}[R_f(Y;\beta_0,\hat{\psi})] - \hat{\psi}^{YX}[R_f(Y;\beta_0,\hat{\psi})]\right\|\right)$$

$$= O_P\left(\left\|\psi^X - \hat{\psi}^X\right\|^2\right) + O_P\left(\left\|\psi^X - \hat{\psi}^X\right\|\left\|\psi^{YX}[H(Y)] - \hat{\psi}^{YX}[H(Y)]\right\|\right)$$

$$= \sum_z O_P\left(\left\|\hat{\xi}_x(z,W) - \xi_x(z,W)\right\|^2 + \left\|\hat{\xi}_x(z,W) - \xi_x(z,W)\right\|\left\|\theta(x,z,W)[H(Y)] - \hat{\theta}(x,z,W)[H(Y)]\right\|\right),$$

where the equalities can be shown using the standard computation and the positivity assumption.

Similarly we assume, for any $x, z$,

$$O_P\left(\left\|\theta(x,z,W)[H(Y)] - \hat{\theta}(x,z,W)[H(Y)]\right\|\left\|\theta(x,1-z,W)[H(Y)] - \hat{\theta}(x,1-z,W)[H(Y)]\right\|\right)$$

$$= \sum_{z\in\{0,1\}} O_P\left(\left\|\theta(x,z,W)[H(Y)] - \hat{\theta}(x,z,W)[H(Y)]\right\|^2\right).$$

We have

$$O_P\left(\left\|\hat{\psi}[H(Y)] - \psi[H(Y)]\right\|^2\right)$$

$$= O_P\left(\left\|\psi^{\hat{Y}X}[H(Y)] - \psi^{YX}[H(Y)] + \hat{\psi}^X - \psi^X\right\|^2\right)$$

$$= O_P\left(\left\|\psi^{\hat{Y}X}[H(Y)] - \psi^{YX}[H(Y)]\right\|^2 + \left\|\hat{\psi}^X - \psi^X\right\|^2 + \left\|\psi^{\hat{Y}X}[H(Y)] - \psi^{YX}[H(Y)]\right\|\left\|\hat{\psi}^X - \psi^X\right\|\right)$$

$$= \sum_{z\in\{0,1\}} O_P\left(\left\|\theta(x,z,W)[H(Y)] - \hat{\theta}(x,z,W)[H(Y)]\right\|^2\right) + \sum_{z\in\{0,1\}} O_P\left(\left\|\xi_x(z,W) - \hat{\xi}_x(z,W)\right\|^2\right)$$

$$+ \sum_{z\in\{0,1\}} O_P\left(\left\|\theta(x,z,W)[H(Y)] - \hat{\theta}(x,z,W)[H(Y)]\right\|\left\|\xi_x(z,W) - \hat{\xi}_x(z,W)\right\|\right).$$

Finally

$$\text{Eq. (B.14)} = \sum_z O_P\left(\|\hat{\pi}_z(W) - \pi_z(W)\|\left\{\left\|\hat{\theta}(x,z,W)[H(Y)] - \theta(x,z,W)[H(Y)]\right\| + \left\|\hat{\xi}_x(z,W) - \xi_x(z,W)\right\|\right\}\right)$$

$$+ \sum_z O_P\left(\left\|\hat{\xi}_x(z,W) - \xi_x(z,W)\right\|^2 + \left\|\theta(x,z,W) - \hat{\theta}(x,z,W)\right\|^2\right)$$

$$+ \sum_z O_P\left(\left\|\hat{\xi}_x(z,W) - \xi_x(z,W)\right\|\left\|\theta(x,z,W) - \hat{\theta}(x,z,W)\right\|\right). \tag{B.31}$$

Therefore, with Eq. (B.13), the following holds

$$\widehat{\beta} - \beta_0 = -M^{-1}\mathbb{E}_{\mathcal{D}}\left[\phi_m(\mathbf{V};\beta_0,\psi_0,\eta_0)\right] + \text{Eq. (B.31)} + o_P(n^{-1/2}),$$

where Eq. (B.31) $= R_2^m$.

$\square$

**Corollary 4** (Restated Corol. 4)**.** *If nuisances $\{\hat{\pi}, \hat{\xi}, \hat{\theta}\}$ converges at $n^{-1/4}$ rate, then the target estimator $\hat{\beta}$ converges to $\beta_0$ at $\sqrt{n}$-rate.*

*Proof.* If all nuisances converge at $n^{-1/4}$ rate, then the $R_2^m$ term in Thm. 3 converges at $n^{-1/2}$ rate. Also, $\mathbb{E}_{\mathcal{D}}\left[\phi_m(\beta_0;(\psi_0,\eta_0))\right]$ converges in distribution to $N(0, \mathrm{var}(\phi_m(\beta_0,(\psi_0,\eta_0))))$ at $\sqrt{n}$-rate. So $\hat{\beta}$ converges to $\beta_0$ at $\sqrt{n}$-rate by Thm. 3. $\qquad\square$

## C  Details of empirical applications

### C.1  Data generating processes for synthetic datasets

The following structural equations are used for all four data generating processes in Fig. 2:

$$U \sim N(0,1)$$
$$f_W(U) = 2U - 1 + \epsilon_W, \text{ where } \epsilon_W \sim N(0,1)$$
$$f_Z(W) = \mathbb{1}\left(0.25W + \epsilon_Z > 0\right), \text{ where } \epsilon_Z \sim N(0,1)$$
$$f_X(W,Z,U) = \mathbb{1}\left(Z + 0.25*W + 0.25*U + \epsilon_X > 0.5\right)\cdot(1-Z) + Z, \text{ where } \epsilon_X \sim N(0,1).$$

With such data generating process, $X_{Z=1} \geq X_{Z=0}$ is satisfied. We will denote four figures in Fig. 2 as Fig. 2(a,b,c,d). For Fig. 2a,

$$f_Y(W,X,U) = 0.6501(W\cdot(2X-1) + 2U + 0.374).$$

For Fig. 2b,

$$f_Y(W,X,U) = 0.9515(2X - 1 + W) + 0.8(-2X + 1 + U) + WU + 0.082.$$

For Fig. 2c,

$$f_Y(W,X,U) = 1.0854\mathbb{1}\left(W < 0\right)(2X - 1 + 0.1U) + \mathbb{1}\left(0 \leq W < 1\right)(-2X + 1 + 0.1U)$$
$$+ 1.0854 \cdot 0.9163\left(\mathbb{1}\left(W \geq 1\right)(-3(2X-1) + 0.2U + 0.3) - 0.122\right)$$

For Fig. 2d,

$$f_Y(W,X,U) = 0.7865 \cdot 1.0628 \cdot \mathbb{1}\left(W < -1\right)(-0.8(2X-1) + 0.1U) + \mathbb{1}\left(-1 \leq W < 0\right)(-2(2X-1) + 0.1U)$$
$$+ 0.7865 \cdot 1.0628 \cdot \left(\mathbb{1}\left(0 \leq W < 1\right)(2(2X-1) + 0.2U) + \mathbb{1}\left(W > 1\right)(0.5(2X-1) + 0.2U) + 0.0525\right)$$
$$+ 1.0628 \cdot 0.104$$

### C.2  Application to 401(k) data

We use the 401(k) dataset that is initially introduced by [2]. Specifically, we used the version of the data named 'The Woodridge Data Set [67]' originally entitled '401ksu.dta' in STATA format (available in `https://www.stata.com/texts/eacsap/`). In the dataset, we used `nettfa` (net financial asset in \$1000) as $Y$, `p401k` (participation in 401(k), participation = 1) as $X$, `e401k` (eligibility for 401(k), eligible = 1) as $Z$, and $W = \{W_1, W_2, W_3, W_4, W_5\} = \{$`agesq, fsize, male, marr, incsq`$\}$, where `agesq` means the square of the age, `fsize` the family size, `male` the gender (male = 1), `marr` the marital status (married = 1) and `incsq` the square of the income.

### C.3  Density plots illustrating uncertainty

In this section, we present the density plots corresponding to Figs. (3,4) illustrating uncertainty of the results. The same data generating processes as used for Figs. (3,4) are leveraged.

**Synthetic dataset.**  To represent the uncertainty, we generate 100 synthetic datasets $\{\mathcal{D}_k\}_{k=1}^{100}$, each of which has $N = 50000$ samples (i.e., $|\mathcal{D}_k| = 50000$), from the same data generating process used for the simulation for Fig. 3. After learning the density estimation with $\mathcal{D}_k$, we obtain a vector of density values $(p_1^k, p_2^k, \cdots, p_m^k)$ at $m$ equi-spaced points for each method ('Moment', 'MLTE', 'Kernel-smoothing', 'KLTE'). For the model-based approach (Moment, MLTE), $m$ is set to 1000. For the kernel-based approach (Kernel-smoothing, KLTE), $m$ is set to 25. For each {model, kernel}-based approach, we have estimates of the density in the form of a matrix $\{(p_1^k, \cdots, p_m^k)\}_{k=1}^{100}$. Let $p_{\mathrm{avg},i}$ denote the average of $\{p_i^k\}_{k=1}^{100}$. Let $\sigma_i$ denote the standard deviation of $\{p_i^k\}_{k=1}^{100}$. Then, we take

$$\mathbf{p}_{\mathrm{avg}} \equiv \{p_{\mathrm{avg},i}\}_{i=1}^m$$
$$\mathbf{p}_{\mathrm{upper}} \equiv \{p_{\mathrm{avg},i} + \sigma_i\}_{i=1}^m$$
$$\mathbf{p}_{\mathrm{lower}} \equiv \{p_{\mathrm{avg},i} - \sigma_i\}_{i=1}^m.$$

A set of density plots corresponding to Fig. 3 with uncertainty information in Fig. C.5. For each density estimate in Fig. C.5, the middle dark-colored dotted line shows $\mathbf{p}_{\text{avg}}$, and the above,below light-colored solid line shows $\mathbf{p}_{\text{upper}}$, $\mathbf{p}_{\text{lower}}$ respectively.

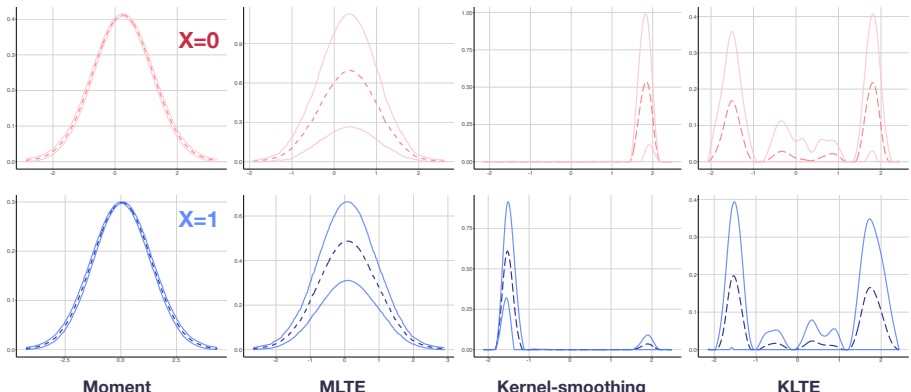

Figure C.5: LTE estimation with a synthetic dataset. The middle dark-colored dotted line denotes $\mathbf{p}_{\text{avg}}$, and the upper and lower light-colored solid lines represents $\mathbf{p}_{\text{upper}}$, $\mathbf{p}_{\text{lower}}$, respectively.

**Application to 401(k) data.** To represent the uncertainty, we randomly resample (with replacement) the dataset from the original dataset $\mathcal{D}$, where the $k$th regenerated dataset is denoted $\mathcal{D}_k$. We conducted this data regeneration process for 100 times and have $\{\mathcal{D}_k\}_{k=1}^{100}$. After learning the density estimation with $\mathcal{D}_k$, we obtain a vector of density values $(p_1^k, p_2^k, \cdots, p_m^k)$ at $m$ equi-spaced points for each method ('Moment', 'MLTE', 'Kernel-smoothing', 'KLTE'). For the model-based approach (Moment, MLTE), $m$ is set to 1000. For the kernel-based approach (Kernel-smoothing, KLTE), $m$ is set to 25. For each {model, kernel}-based approach, we have estimates of the density in the form of a matrix $\{(p_1^k, \cdots, p_m^k)\}_{k=1}^{100}$. Let $p_{\text{avg},i}$ denote the average of $\{p_i^k\}_{k=1}^{100}$. Let $\sigma_i$ denote the standard deviation of $\{p_i^k\}_{k=1}^{100}$. Then, we take $\mathbf{p}_{\text{avg}} \equiv \{p_{\text{avg},i}\}_{i=1}^{m}$, $\mathbf{p}_{\text{upper}} \equiv \{p_{\text{avg},i} + \sigma_i\}_{i=1}^{m}$ and $\mathbf{p}_{\text{lower}} \equiv \{p_{\text{avg},i} - \sigma_i\}_{i=1}^{m}$.

A set of density plots corresponding to Fig. 3 with uncertainty information in Fig. C.6. For each density estimate in Fig. C.6, the middle dark-colored dotted line shows $\mathbf{p}_{\text{avg}}$, and the above,below light-colored solid line shows $\mathbf{p}_{\text{upper}}$, $\mathbf{p}_{\text{lower}}$ respectively.

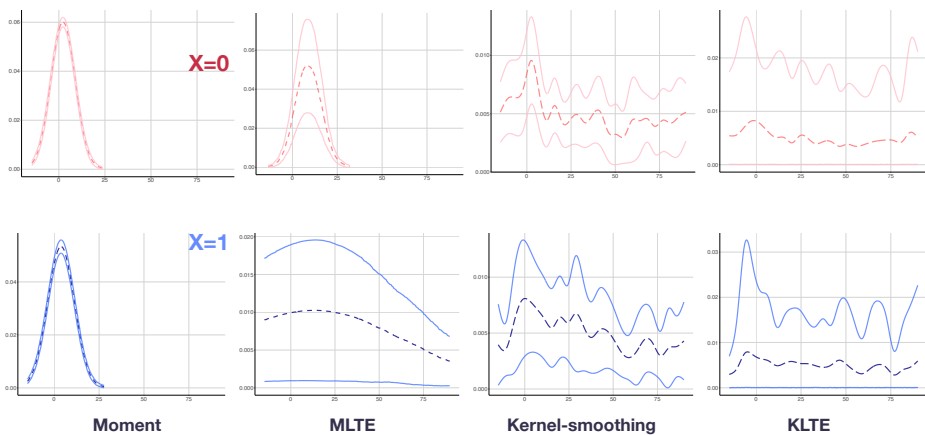

Figure C.6: LTE of 401(k) participation $(X)$ on net financial asset $(Y)$. The middle dark-colored dotted line denotes $\mathbf{p}_{\text{avg}}$, and the upper and lower light-colored solid lines represents $\mathbf{p}_{\text{upper}}$, $\mathbf{p}_{\text{lower}}$, respectively.