# OpenReview forum: "Double Machine Learning Density Estimation for Local Treatment Effects with Instruments"
_NeurIPS.cc/2021/Conference — NeurIPS 2021 Spotlight_

### Official Review · Reviewer_86MR · 2021-07-15

**Rating:** 7
**Confidence:** 4

**Summary:**

This paper proposes two methods for estimating the density of the outcome variable for the compliers in the potential outcomes framework. The first is a kernel-smoothing based approach, while the second is a model-based approach, and in both cases, the double machine learning approach based on data splitting is employed. The nature of the contribution seems mostly theoretical, with a lot of effort into proving the convergence rates.

**Limitations And Societal Impact:**

The nature of the work is mostly theoretical, and therefore I deem it not necessary to consider societal impact.

**Main Review:**

I thank the authors for their work put into this submission. I tried to do it justice with my review; however, given time constraints, and my lack of prior knowledge on some of the previous work, there are likely to be misunderstandings on my part, and I do apologise in advance for those.

I believe looking at distributional aspects of treatment effect beyond the mean is an important issue, and looking at the density is one interesting way of doing it. Since considering distributional aspects of treatment effects have received a lot of attention (by looking at cumulative distribution functions or quantiles, as the authors mention, but also by looking at kernel mean embeddings; see [Counterfactual mean embeddings, Muandet et al., 2020] or [Kernel methods for policy evaluation, Singh et al., 2020], for example), it is quite surprising that this paper is among the first to look at densities. This paper also applies ideas of double machine learning, in which I see value. Overall, I think this paper should be seen as a natural follow-up to existing ideas, rather than a really original piece of work.

The paper is generally fairly well-written, although I think the clarity of the development could be improved. For example, in Section 3, I think it would be helpful to postulate explicitly what “influence function”, “moment condition” and “Neyman orthogonal score” refer to, and why they are used, in clear mathematical terms (rather than in brackets and in footnotes, as currently is written).

The following are some specific points that arose during my review. Most of them are very minor typos, but I think some of the points could be significant, but it is also likely that I am mistaken about them. I will give an overall score of 6 for now, although if all the points are adequately addressed, I would be happy to raise the score.

1) line 25: it will help -> will help
2) line 70: ‘debaisedness’ -> ‘debiasedness’
3) lines 101-104: Just a matter of notation, but I would prefer the distinction between the function and its evaluation to be clear. So I would write “… density p and a function f, … || f || = sqrt{E_P[(f(X))^2]}, … if || hat{f} - f ||=O_P(1/r_n).
4) footnote 2: which -> whose
5) lines 142-147: The kernel K takes y in R^d as argument, so in the examples of kernels given, instead of u^2, I think you would need || u ||^2. Also, For the same reason as above, I think it would be better to write || K || on line 144 rather than || K(y) ||; we’re considering the norm of the function K, rather than the norm of the real number K(y).
6) displayed equation after line 569: on the first line, shouldn’t phi_X[f(Y)]  be just phi_X? And on the second line, shouldn’t psi^X[f(Y)] be just psi^X? On the second, third and fourth lines, shouldn’t V_X[f(Y)] just V_X, or V_X({pi,xi})?
Also, I have a question on this proof. It seems that the influence function of a quotient A/B is given as influence(A)/B - influence(B)A/B^2, just like the quotient rule for differentiation. Is this obvious? I would have preferred it if there was some explanation / lemma as to why this is true.
7) line 577: is is -> is
8) The condition on line 595 is presumably to satisfy the condition on line 554 in Lemma S.1, but there, the nuisance is also the estimate hat{eta}, whereas here the true eta is used. Is the condition still satisfied for hat{eta}, which I presume is what is required?
9) Is M=-1 obvious? Could you please write this out?
10) displayed equations after line 606: the 0 expression added could be written so that the + and the - are swapped; this would be consistent with the next line.
11) displayed equations after line 607: in the last term on the third line, isn’t psi_h missing? Also, throughout, I think I would have preferred psi__h being written as psi_h(y), since it is dependent on a particular choice of y, and K_{h,y} is always written with explicit dependence on y. But it might clutter up the notation, so I’ll leave the decision to the authors.
12) line 616: psi_h seems to be missing here too, and the displayed equations that follow. This actually seems like it will make a difference to the rate?
13) displayed equations after line 620, first line: phi should be phi_m.
14) displayed equations after line 630: presumably, psi_{YX}+R_{YX}=hat{V}_{YX}, which means (ignoring the fact that the right-hand side is random and the left-hand side is random, since expectation is taken in the end anyway) the expectation of V_{YX} is psi_{YX}? and psi_X+R_X=hat{V}_X?
14) line 632: shouldn’t hat{V}_{YX}, V_{YX}, hat{V}_X and V_X be their expectations? Otherwise a random element is equal to a non-random element, since R_{YX} and R_X are not random.
15) line 152: in the definition of theta, there is no dependence on z on the right-hand side. Shouldn’t x in the conditioning be z?
16) line 153: there is a redundant ).

**Time Spent Reviewing:**

15

---

> ### Author Response · Authors · 2021-08-10
> **Response to reviewer 86MR**
>
> Thank you for the valuable feedback and detailed comments on the proofs. We will definitely use your comments to improve the paper. Also, we appreciate you sharing the references on kernel mean embeddings (Muandet et al. 2020 & Singh et al. 2020), they seem quite interesting and we will read and acknowledge in the paper.
>
> > I think it would be helpful to postulate explicitly what “influence function”, “moment condition” and “Neyman orthogonal score” refer to, and why they are used, in clear mathematical terms (rather than in brackets and in footnotes, as currently is written).
>
> We struggle to fit all the content of the paper in the allotted space, but will use the additional extra page to add these definitions (if the paper is accepted). Thank you for the suggestion.
>
> >The following are some specific points that arose during my review.
>
> We really appreciate your careful reading and checking of the proofs. We will fix the typos in Points 1-4, 7, 10, 11, 13, 16, 17 accordingly. The remaining points are addressed in the following.
>
> > 5. lines 142-147: The kernel K takes y in R^d as argument, so in the examples of kernels given, instead of u^2, I think you would need || u ||^2. Also, For the same reason as above, I think it would be better to write || K || on line 144 rather than || K(y) ||; we’re considering the norm of the function K, rather than the norm of the real number K(y).
>
> The product kernel $K_{h, y}$ takes $y$ in $\mathbb{R}^d$ as argument, while the kernel $K$ takes $y_j \in \mathbb{R}$ as argument. Agree on the norm notation.
>
> > 6. displayed equation after line 569: on the first line, shouldn’t phi_X[f(Y)] be just phi_X? And on the second line, shouldn’t psi^X[f(Y)] be just psi^X? On the second, third and fourth lines, shouldn’t V_X[f(Y)] just V_X, or V_X({pi,xi})?  Also, I have a question on this proof. It seems that the influence function of a quotient A/B is given as influence(A)/B - influence(B)A/B^2, just like the quotient rule for differentiation. Is this obvious? I would have preferred it if there was some explanation / lemma as to why this is true.
>
> Thanks for the catch, these glitches will be fixed. The influence function of a target estimand in the form of A/B is given by applying the quotient rule because the influence function is defined as a pathwise derivative. We will try to add further explanation around this line.
>
> > 8. The condition on line 595 is presumably to satisfy the condition on line 554 in Lemma S.1, but there, the nuisance is also the estimate hat{eta}, whereas here the true eta is used. Is the condition still satisfied for hat{eta}, which I presume is what is required?
>
> The $\eta$ in line 595 meant arbitrary nuisances, and specifically for the KLTE estimator it will be  \hat{\eta}. With $\hat{\eta}$, it’s the case that the KLTE $\hat{\psi_2}$ is satisfying the condition in line 554 $\mathbb{E}_{\mathcal{D}}[ \varphi( \hat{\psi}_h, \hat{\eta} )  ] = 0$; i.e., the condition in line 554 is satisfied with $\hat{\eta}$.
>
> > 9. Is M=-1 obvious? Could you please write this out?
>
> $M$ is a derivative of $\mathbb{E}[\varphi(\psi_h, \eta_0)]$ w.r.t. an arbitrary $\psi_h$ evaluated at the true $\psi_h$. We first note that $\mathbb{E}[\varphi(\psi_h, \eta_0)] = {\psi_{YX}/\psi_{X}}  - \psi_h$. Taking a derivative w.r.t. arbitrary $\psi_h$ is -1. Evaluating it at the true $\psi_h$ also gives the derivative as $-1$. Therefore, $M = -1$. We’ll add this more explicitly in the text.
>
>
> > 12. line 616: psi_h seems to be missing here too, and the displayed equations that follow. This actually seems like it will make a difference to the rate?
>
> The rate still remains the same, because $\mathbb{E}[\phi_{YX}[K_{h,y}(Y)] \cdot \phi_{X}] = O(h^{-d/2}), \psi_h = O(h^{-d/2})$ as shown in line 615. This makes the term $\psi_h  \mathbb{E}[ \phi_{YX}[K_{h,y}(Y)] \cdot \phi_{X}] = O(h^{-d})$, which still leads to the rate in line 619.
>
> > 14. displayed equations after line 630: presumably, psi_{YX}+R_{YX}=hat{V}{YX}, which means (ignoring the fact that the right-hand side is random and the left-hand side is random, since expectation is taken in the end anyway) the expectation of V{YX} is psi_{YX}? and psi_X+R_X=hat{V}_X?
>
> Yes. As pointed out, these equalities hold in the expectation. Specifically, $\mathbb{E}[ \psi_{YX} + R_{YX} ] = \mathbb{E}[ \psi_{YX} + \hat{V_{YX}} - V_{YX} ] = \mathbb{E}[ \hat{V_{YX}}]$, since $\mathbb{E}[V_{YX}] = \psi_{YX}$. Similarly, $\mathbb{E}[ \psi_{X} + R_X ] = E[ \psi_{X} + \hat{V_X} - V_{X} ] = E[ \hat{V_{X}} ]$ holds.
>
>
> > 15. line 632: shouldn’t hat{V}{YX}, V{YX}, hat{V}X and V_X be their expectations? Otherwise a random element is equal to a non-random element, since R{YX} and R_X are not random.
>
> Yes, they should be in the expectation.

---

### Official Review · Reviewer_A9r3 · 2021-07-16

**Rating:** 7
**Confidence:** 3

**Summary:**

The most common problem in causal effect estimation is that of estimating average effects over populations. However, to get to individual level causal effects, one has to estimate the full *distribution* or densities in the continuous outcome case. The authors are estimating such a density for compliers in the IV setting. Two density estimation strategies are provided i) kernel-based, ii) model-based. Essentially the core contribution is that in the identifiable setting (under monotonicity), authors have extended density estimation methods to the IV setting for estimating density of causal effects for compliers. The key methodology used is double machine learning which has extensive precedent for designing robust estimators for causal inference problems. I think the contribution is valuable to the community and toward making causality realistic and using it for real-world applications.

**Ethical Concerns:**

I do not see ethical concerns with the paper.

**Limitations And Societal Impact:**

Major limitations I think are lack of clarity in places where certain details are hidden and leaves the reader to fill gaps. I recommend fixing this. There are no negative societal impacts of this work.

**Main Review:**

The technical novelty of model-based part is sort of unclear (authors are proposing a robust estimator that has previously only been derived in the fully observed confounding case). However the contributions in the kernel smoothing part is unclear in comparison to cited work. Is it also that previously kernel based approaches have only been proposed for say, the bow-graph and common settings and not the IV setting considered here?

The authors have not discussed any details or insights about model selection. Atleast this is less of an issue for kernel based methods, maybe one can tweak kernel parameters till estimation errors look reasonable. However the parametrizations in the model based estimation seems to need significant amount of domain knowledge. How should one choose the family of g. I understand it might be hard but please include a few comments reagarding practicality of proposed methods.


The estimand in line 149 should be more clearly derived.

Lemma (S.1) appendix. Define Donsker (Link shows up only later so best move to S.1)

Line 598 in appendix: Can the authors please expand the derivation they have used for asymptotic analysis. It has taken a long time to actually verify whether this is correct. If the authors can update the appendix, I am happy to review again.

I have only made cursory checks to proofs in Theorem 1 and 3 but suggests they are reasonable.
Minor:
1. Line 77: confusing line - As shown in Fig. 2, however, the average is sometimes insufficient to capture the treatment effects on distributions. -> to capture the distribution of treatment effects(?)


**Time Spent Reviewing:**

3-4 hours

---

> ### Author Response · Authors · 2021-08-10
> **Response to reviewer A9r3**
>
> Thank you for the valuable feedback. We will use your comments to improve the paper. Please, find below further clarification on the points raised.
>
> > "The technical novelty of model-based part is sort of unclear (authors are proposing a robust estimator that has previously only been derived in the fully observed confounding case). However the contributions in the kernel smoothing part is unclear in comparison to cited work. Is it also that previously kernel based approaches have only been proposed for say, the bow-graph and common settings and not the IV setting considered here?”
>
> The literature on estimating the density of treatment effects has focused on ignorability/backdoor settings (i.e., the fully observed confounding case)  for both model-based and kernel-smoothing methods (discussed in lines 87-96). It is the case that  DML-style density estimators have not been developed in the IV setting with kernel smoothing methods, nor with model-based methods. We feel the work is technically novel because applying the model-based and kernel-smoothing strategies to the IV setting is technically nontrivial, including the derivation of influence functions, convergence analyses, and proving doubly robustness & debiasedness.
>
> > “The authors have not discussed any details or insights about model selection. …  I understand it might be hard but please include a few comments reagarding practicality of proposed methods.”
>
> Thanks for the suggestion, we will add a discussion regarding model selection. Roughly speaking, one may use data to choose among a set of candidate families for the model-based methods.
>
> > “The estimand in line 149 should be more clearly derived.”
>
> We will add a detailed derivation in the appendix.
>
> > “Lemma (S.1) appendix. Define Donsker (Link shows up only later so best move to S.1)”
>
> Thanks, we’ll move the definition to S.1.
>
> > “Line 598 in appendix: Can the authors please expand the derivation they have used for asymptotic analysis. It has taken a long time to actually verify whether this is correct. If the authors can update the appendix, I am happy to review again.”
>
> Sorry for the inconvenience. We will carefully go through proofs to expand the derivation to provide more explicit details. (As we understand, updating the submission is not allowed at this point.) Please note that the equation in line 598 follows directly from Lemma S.1.
>
> > Line 77: confusing line - As shown in Fig. 2, however, the average is sometimes insufficient to capture the treatment effects on distributions. -> to capture the distribution of treatment effects(?)
>
> A: This passage should read as “to capture the effects of the treatment on the distributions of outcomes”.

---

### Official Review · Reviewer_PVR7 · 2021-07-17

**Rating:** 7
**Confidence:** 2

**Summary:**

The authors propose kernel-smoothing-based and model-based approaches for estimating the LTE density in the presence of instruments. They derive Neyman orthogonal scores and construct the corresponding DML estimators (KLTE and MLTE), that exhibit debiasedness property. They experiment on synthetic and real datasets.

**Limitations And Societal Impact:**

The authors should further discuss the impact of potential defiers (Assumption 1) in the algorithm.


**Main Review:**

Overall, the paper is well-written and easy to follow.

The theoretical results seem thorough and address multiple aspects of bias and consistency of the proposed algorithms. Derivations of the Neyman orthogonality scores for the two cases are interesting.

In lines 32-33, it is unclear if the argument in (ii) holds in reality. It is possible that the availability of 401k depends on the confounders too.

The experiments are inadequate. The authors only report plots without any measure of uncertainty. I encourage the authors to report the actual accuracy numbers with a measure of uncertainty so that we can measure the significance of the improvements.

**Time Spent Reviewing:**

2

---

> ### Author Response · Authors · 2021-08-10
> **Response to reviewer PVR7**
>
> Thank you for the valuable feedback.
>
> > “In lines 32-33, it is unclear if the argument in (ii) holds in reality. It is possible that the availability of 401k depends on the confounders too.”
>
> We assume you meant (iii) since it’s about confounders. This is a classic example from the literature where this particular unconfoundedness statement is assumed to hold, e.g., see [2, 15, 41]. The idea is that eligibility ($Z$) is largely decided by known eligibility criteria, which is part of $W$. In some way, we are using it as a benchmark to our method, but our setting is agonistic to this particular application.
>
>
> > “The experiments are inadequate. The authors only report plots without any measure of uncertainty.
>
> We appreciate your suggestions on improving the experiments and will do so. (We’ve done adding shades around the curves in the plots to represent uncertainty, but they might not be visually appealing.) The plots in Figs. 3 and 4 clearly show the important characteristics of densities (such as modes, locations, scale) output by different estimators (Fig. 3 also shows the ground-truth density). We will add plots with shades representing uncertainty to the appendix in the revised manuscript, thank you.
>
> > (Under Limitations & Societal impact) “The authors should further discuss the impact of potential defiers (Assumption 1) in the algorithm.”
>
> Thanks for the suggestion. The work is limited to settings where Assumption 1 holds; that is, only individuals who were offered the treatment may have access to it (i.e., monotonicity holds). We note that this assumption is quite common and believed to hold in many settings [2, 3, 28]. Still, we will include some references to the literature where sensitivity analyses w.r.t the potential defiers to the estimates is conducted.

---

### Official Review · Reviewer_GUHn · 2021-07-20

**Rating:** 7
**Confidence:** 3

**Summary:**

This paper deals with the problem of the estimation of the local treatment effect (LTE) which is measure of the causal impact of a treatment. To this extent the authors develop kernel-smoothing and model-based methodologies to recover this quantity from the data.
The theoretical properties of the machinery are developed and the authors show that the estimator of the LTE is consistent.
Two applications of the proposed methodology showcase the technique.

**Ethical Concerns:**

this paper does not raise any concerns.

**Limitations And Societal Impact:**

this paper does not raise any concerns.

**Main Review:**

This is a great paper which deals with causal inference based on machine learning methods. The authors know the literature very well. The paper is well written. The paper has both interesting theoretical results as well as a nice empirical application. This paper would make a great contribution to the program.

**Time Spent Reviewing:**

4

---

> ### Author Response · Authors · 2021-08-10
> **Response to reviewer GUHn**
>
> Thank you for your feedback and comments, much appreciated.

---

### Decision · Program_Chairs · 2021-09-27

**Decision:**

Accept (Spotlight)

**Comment:**

The reviewers unanimously found your paper to be compelling. I expect you will make the changes discussed during the review process.